# ST-GCond: Self-supervised and Transferable Graph Dataset Condensation

**Beining Yang[1], Qingyun Sun[2]\*, Cheng Ji[2], Xingcheng Fu[3], Jianxin Li[2],**
[1]Univerisity of Edinburgh    [2]Beihang University    [3]Guangxi Normal University
`B.Yang-32@sms.ed.ac.uk, sunqy@buaa.edu.cn`

## Abstract

The increasing scale of graph datasets significantly enhances deep learning models but also presents substantial training challenges. Graph dataset condensation has emerged to condense large datasets into smaller yet informative ones that maintain similar test performance. However, these methods strictly require downstream usage to match the original dataset and task, leading to failures in cross-task and cross-dataset scenarios. To address such cross-task and cross-dataset challenges, we propose a novel **S**elf-supervised and **T**ransferable **G**raph dataset **Cond**ensation method named ST-GCond, providing effective and transferable condensed datasets. Specifically, for cross-task challenge, we propose a task-disentangled meta optimization strategy to adaptively update the condensed graph according to the task relevance, encouraging information preservation for various tasks. For cross-dataset challenge, we propose a multi-teacher self-supervised optimization strategy to incorporate auxiliary self-supervised tasks to inject universal knowledge into the condensed graph. Additionally, we incorporate mutual information guided joint condensation mitigating the potential conflicts and ensure the condensing stability. Experiments on both node-level and graph-level datasets show that ST-GCond outperforms existing methods by $2.5\% \sim 18.7\%$ in all cross-task and cross-dataset scenarios, and also achieves state-of-the-art performance on 5 out of 6 datasets in the single dataset and task scenario.

## 1 Introduction

Dataset plays an essential role in contemporary machine learning research (Liu et al., 2024; Ghorbani et al., 2022). It is also widely believed that there is a scaling law between dataset size and the power of deep learning models (Kaplan et al., 2020). Graph datasets, in particular, contain vast amounts of relations and entities that represent complex interactions, such as those found in social networks, molecules, and recommender systems. These datasets serve as the foundation for many powerful models designed to support various analytical applications (Schroff et al., 2015; Wu et al., 2019; Battaglia et al., 2018; Sun et al., 2022). However, large-scale graph datasets also present significant challenges in terms of storage, processing, and computational resources. On one hand, specific applications like neural architecture search (Zhang et al., 2022; Guan et al., 2022) and continual learning (Choi et al., 2024) require repetitive training on datasets, resulting in significant computational costs. On the other hand, for users with limited computational resources, training on large-scale graph datasets can be exceedingly time-consuming or even impractical. Recently, several *graph dataset condensation* methods (Xu et al., 2024; Sun et al., 2024) have been proposed to address this and show remarkable success in preserving essential information with extremely small datasets.

However, as shown in Fig. 1a, all existing graph condensation methods are designed to condense a specific dataset tailored for single dataset and task. This limitation is incompatible with real-world requirements, where users may encounter new or user-defined data and tasks that significantly differ from those condensed graph datasets derived by open-access datasets. Unlike applying applications to a few public graph datasets, such scenarios are much more common and challenging. Though there exist other lines of studies (e.g., graph transfer learning (Zhuang et al., 2020), graph meta learning (Mandal et al., 2022), and graph foundation models(Liu et al., 2023a)) that could also address

---

\*Corresponding author.

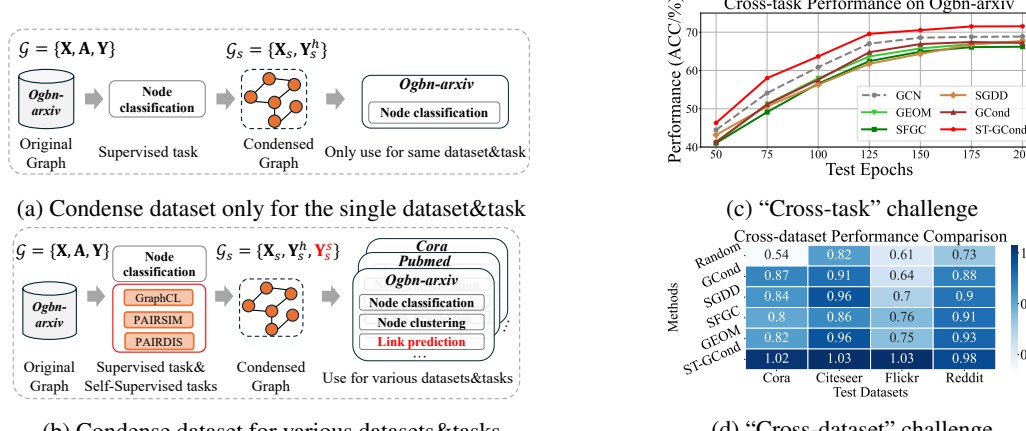

Figure 1: We illustrate the pipeline of existing graph condensation methods (Fig. 1a), which are designed for a single dataset and task. We enhance these methods to make the condensed graph transferable to various downstream needs (Fig. 1b). We achieve faster and more effective performance in both cross-task (Fig. 1c) and cross-dataset (Fig. 1d) scenarios.

this issue, they require fixed model architecture design or complex training strategies, and relatively huge computational resources. Therefore, it is natural to ask a question: *Can the condensed graph dataset be used to train models for various tasks and upon various datasets?*

To simplify the discussion, we extend the concept of model transferability to **data-level transferability**, which evaluates how well the condensed dataset, when used to train a model, enables the model to transfer effectively to new datasets or new tasks. Ideally, enhanced transferability would allow users to train models more efficiently, achieving faster convergence and better results when applied to their own data or task (Fig. 1b). However, we empirically find there are two main challenges for addressing the problem of transferability on the current graph dataset condensation methods.

**Challenge ❶: Efficient and Fast Cross-task Adaptation.** In this context, a "task" refers to either changes in task types or changes in label sets. We provide a detailed explanation of these distinctions and present experiments for both scenarios in Sec. 4.2. Here, we focus on an example of label set changes, as illustrated in Fig. 1c. Using the Ogbn-arxiv dataset, we apply condensation based on only half of the supervised data (i.e., 20 classes). A 2-layer GCN model, trained on this condensed graph, is then tested on the remaining 20 classes by fine-tuning the final linear layer. We show the performance of various methods over fine-tuning epochs (test epochs), with the gray lines representing the results of a naive 2-layer GCN model. Existing methods underperform by 4.2% compared to the ground truth averagely, highlighting the difficulty of cross-task transferability.

**Challenge ❷: Cross-dataset Universal Information Preservation.** We further investigate cross-dataset scenarios in the Fig. 1d, where a 2-layer GCN model trained on condensed Ogbn-arxiv data is generalized to four different datasets by replacing the final layer with a new linear layer. To illustrate the differences more clearly, we normalize each result to the performance of the naive 2-layer GCN model. In this context, Random refers to a model that was not trained on the condensed dataset but instead fine-tunes only the final layer. While current methods outperform the Random baseline, they still fail to surpass the performance of a simple GCN model, remaining below 1.0 ratio.

To address the above challenges, we propose a **S**elf-supervised and **T**ransferable **G**raph dataset **Cond**ensation framework named **ST-GCond**. Unlike existing methods focused on condensing a specific dataset for a single task, ST-GCond is designed to encourage the condensed graph to preserve the most general and informative patterns, leading to better transferability across tasks and datasets. Specifically, to achieve better cross-task transferability, the ST-GCond utilizes *a task-disentangled meta optimization strategy* to adaptively update the condensed graph according to the task relevance. To achieve better cross-dataset transferability, the ST-GCond adopts *a multi-teacher self-supervised optimization strategy*: we pre-train several self-supervised models as teachers and adaptively combine them to extract fundamental information from the original dataset. To align above two strategies for joint condensing, *we introduce a mutual information guided joint condensation strategy*, which

mitigates potential conflicts between supervised and self-supervised tasks. We empirically evaluate our `ST-GCond` on 5 node-level and 5 graph-level real-world graph datasets. For the cross-dataset and cross-task scenario, `ST-GCond` outperforms the existing graph condensation methods 2.5% to 18.7%. In the traditional single dataset and task scenario, `ST-GCond` also gets state-of-art performance on 5 out of 6 datasets, demonstrating its versatility. The contributions are as follows:

- We propose `ST-GCond`, a self-supervised and transferable graph dataset condensation framework. To the best of our knowledge, `ST-GCond` is the first graph condensation method that is designed for cross-dataset and cross-task scenarios.
- `ST-GCond` incorporates the task-disentangled and self-supervised optimization to inject the universal knowledge into the condensed graph, enhancing both cross-task and cross-dataset transferability.
- Extensive experiments on 10 real-world datasets demonstrate that `ST-GCond` enjoys the state-of-art performance on both single task and cross-dataset/cross-task scenarios.

## 2 RELATED WORKS

### 2.1 DATASET DISTILLATION

Dataset distillation (Wang et al., 2018; Bohdal et al., 2020; Cui et al., 2022; Wu et al., 2024) is proposed to significantly reduce the scale of the dataset but ensuring the test performance of the distilled dataset. It is suitable for various applications like continual learning, neural architecture search, and for users who have limited computational resources. However, there is still much more gap between the condensed dataset and the original one. For example, the dataset provides more intrinsic knowledge beyond the human-made labels (Kaplan et al., 2020) and can be transferred to benefit the new dataset's tasks (Noroozi et al., 2018; Zamir et al., 2018). Although `KRR-ST` (Lee et al., 2024) is proposed to distill a transferable dataset fitting this goal, the cost is that the distilled dataset cannot be effectively leveraged by traditional continual learning and neural architecture search applications. Therefore, there is a need to comprehensively fill the dataset gap.

### 2.2 GRAPH DATASET CONDENSATION

Graph data have its unique feature that the samples (nodes) are not independent. Therefore, distilling (condensing) a graph is more difficult for jointly considering the node and structure. Recently, several graph dataset condensation methods (Xu et al., 2024; Gao et al., 2024) have been proposed to achieve better performance on a single dataset and tasks. For instance, `GCond` (Jin et al., 2022c) first introduce the gradient matching method in condensing graph, `SGDD` (Yang et al., 2023) and `SFGC` (Zheng et al., 2023) enhance `GCond` through different ways: further considering the structure and removing the structure. And the most recent `GEOM` (Zhang et al., 2024) following `SFGC` achieves lossless results (even better than solely training a GCN model on the original graph). `DosCond` (Jin et al., 2022b) and `KiDD` (Xu et al., 2023) provide a way to condense the graph classification datasets, while `KiDD` propose to use the Kernel ridge regression to efficiently reduce the condensing time. However, existing methods are still designed for specific datasets and tasks. How to achieve transferable condensation remains under-explored and requires further research.

## 3 SELF-SUPERVISED AND TRANSFERABLE GRAPH DATASET CONDENSATION

### 3.1 OVERALL FRAMEWORK OF ST-GCOND

In this paper, we aim to propose a **transferable graph dataset condensation** method for cross-task and cross dataset scenarios. Given a graph dataset $\mathcal{G} = (\mathbf{X}, \mathbf{A}, \mathbf{Y})$, where $\mathbf{X} \in \mathbb{R}^{N \times d}$ represents the feature matrix, $\mathbf{A} \in \mathbb{R}^{N \times N}$ represents the adjacency matrix, $N$ is the number of nodes, and $d$ denotes the node feature dimension. Our goal is to generate a smaller synthetic graph dataset $\mathcal{G}_s = (\mathbf{X}_s, \mathbf{A}_s, \mathbf{Y}_s^h)$ with $m$ nodes ($m \ll N$), aiming to make any model trained on $\mathcal{G}_s$ achieve similar test performance to a model trained on $\mathcal{G}$ on the same task (defined by the label $\mathbf{Y}$) without lost the generalization ability when transferring such model to the new datasets or tasks. Unlike traditional condensation methods where the condensed graph is restricted to a specific dataset and task, transferable graph dataset condensation aims to expand the usage scope of the condensed graph.

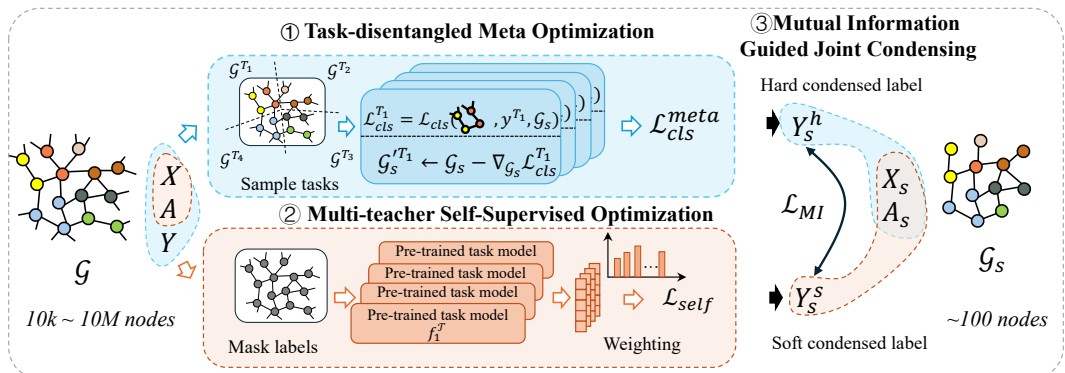

Figure 2: Overall framework of ST-GCond. The original large graph dataset $\mathcal{G}$, consisting of $\mathbf{X}$, $\mathbf{A}$, and $\mathbf{Y}$, is condensed using ① task-disentangled meta-optimization to synthetic hard label $\mathbf{Y}_s^h$ and ② multi-teacher self-supervised optimization to generate soft label $\mathbf{Y}_s^s$, along with their shared $\mathbf{X}_s$, $\mathbf{A}_s$. ③ To avoid conflicts in optimization directions, mutual information-guided joint condensation is employed to serve as the mutual information regulations loss term.

To this end, we propose the **S**elf-supervised and **T**ransferable **G**raph dataset **Cond**ensation framework, termed ST-GCond. As illustrated in Fig. 2, our goal is to condense the original graph dataset $\mathcal{G}$ into a smaller yet informative graph dataset, $\mathcal{G}_s = (\mathbf{A}_s, \mathbf{X}_s, \mathbf{Y}_s^h, \mathbf{Y}_s^s)$. It is worth noting that we include a soft label, $\mathbf{Y}_s^s$, to handle more complex scenarios.

**(1) Task-disentangled Meta Optimization. (Challenge ❶)** We first disentangle the given task (*i.e.*, classes) into $t$ parts, with each part containing $h$ classes ($h > 1$) as sub-task $\{T_i\}_{i=1}^t$. In every epoch, the condensed graph $\mathcal{G}_s$ will first be fast updated through the guide of each sub-task to be the $\{\mathcal{G}_s^{'T_i}\}_{i=1}^t$, then optimize the $\mathcal{G}_s$ through the joint evaluation loss on all sub-tasks. The motivation behind this is to let the condensed graph not only be aware of the difference of the tasks but can quickly adapt to the optimal state of each sub-task. Such process is associated with the hard condensed label $\mathbf{Y}_s^h \in \mathbb{R}^{m \times 1}$.

**(2) Multi-teacher Self-Supervised Optimization. (Challenge ❷)** We leverage self-supervised tasks to extract the universal knowledge of $\mathcal{G}$. Specifically, we load $k$ pre-trained models, each trained on a distinct self-supervised task, denoted as $\{f_1^{\mathcal{T}}(\cdot), f_2^{\mathcal{T}}(\cdot), \cdots, f_k^{\mathcal{T}}(\cdot)\}$. We further propose a synthetic soft label $\mathbf{Y}_s^s \in \mathbb{R}^{m \times d}$ to unify the labels across these tasks. Consequently, the condensed graph is guided by multiple self-supervised tasks, enabling it to acquire more transferable cross-dataset knowledge.

**(3) Mutual Information Guided Joint Condensation. (Resolving the potential conflicts)** Through the discussion of previous literature (Jin et al., 2022a; Fan et al., 2024) and our empirical findings in Tab. 6, we observe that directly combining different self-supervised tasks with the supervised task can lead to unexpected performance drops. Therefore, we propose to constrain the mutual information $I(\mathbf{Y}_s^s; \mathbf{Y}_s^h)$, which adaptively adjusts the weights for each self-supervised task and ensures that the final distribution closely approximates the ideal distribution of $\mathbf{Y}_s^h$, thereby resolving potential conflicts in the joint condensing process.

## 3.2 CROSS-TASK TRANSFERABILITY: TASK-DISENTANGLED META OPTIMIZATION

Existing methods for graph dataset condensation primarily focus on task performance related to the single task and dataset, making the condensed graph only reserve knowledge specific to such a single task, which significantly hampers its transferability to other tasks (see Fig. 1c, Table 4, and Table 5).

Since the actual downstream task information is unknown, preparing an optimal condensed graph tailored for a specific task is impossible. Inspired by MAML (Finn et al., 2017; Zhou et al., 2019), we modify the optimization strategy to search for a "global minimum" across all tasks, enabling the condensed graph to rapidly and effectively adapt to new tasks. Given a distribution over tasks $p(\mathbf{Y})$, we randomly sample label sets $\{y_i\}_{i=1}^t \sim p(\mathbf{Y})$ and induce the sub-task graph $\mathcal{G}^T$. This process is

similar to class-level sampling in `GCond`, but our focus on the task level makes it more suitable for addressing challenge.

**Initialization of the Condensed Graph $\mathcal{G}_s$.** We initialize $\{\mathbf{X}_s, \mathbf{Y}_s^h\}$ by randomly selecting a subset of the original data. Following `GCond`, we use an MLP $g_\phi$ as the structure generator, where $\mathbf{A}_s = (g_\phi(\mathbf{X}_s) - \delta)$, Here, $\delta$ serves as the sparsity parameter to filter out edges with lower weights. Note that $\mathbf{Y}_s^h$ is analogous to the labels $\mathbf{Y} \in \mathbb{R}^{N \times 1}$ of $\mathcal{G}$, encompassing $C$ classes.

**Fast Adaptation and Meta Optimization Strategy.** The key step is simulating the cross-task scenario in the condensing stage with $\mathcal{G}_s$. Following MAML(Finn et al., 2017), we use the meta optimizing strategy, specifically, given the $i$-th sub-task $\mathcal{G}_i^T$, we update a copy of $\mathcal{G}_s$ as follows: $\mathcal{G}_s^{'T_i} \leftarrow \mathcal{G}_s - \nabla_{\mathcal{G}_s} \mathcal{L}_{cls}^{T_i}(\mathcal{G}^{T_i}, \mathcal{G}_s)$. The final optimization for $\mathcal{G}_s$ is then calculated using the following:

$$\mathcal{L}_{cls}^{\text{meta}} = \frac{1}{t} \sum_{i=1}^{t} \mathcal{L}_{cls}(\mathcal{G}^{T_i}, \mathcal{G}_s^{'T_i}). \tag{1}$$

The difference is that the meta-loss term is calculated using the fast-adapted condensed graph, simulating the scenario where users update their models on new data and test on the updated versions.

**Kernel Ridge Regression-Based Condensing Objective.** The term $\mathcal{L}_{cls}$ in Eq. (1) serves as the surrogate condensing objective. For instance, gradient matching (Jin et al., 2022c; Yang et al., 2023) mimics the real gradients occurring in the original dataset, while trajectory matching (Zheng et al., 2023; Zhang et al., 2024) simulates the learning trajectories. In contrast, we adopt the Kernel Ridge Regression (KRR) method (Xu et al., 2023; Wang et al., 2024) to simplify the condensing process, aligning it with the requirements of task-disentangled meta optimization. Formally, $\mathcal{L}_{cls}$ is:

$$\mathcal{L}_{cls}^y \triangleq \min_{\mathcal{G}_s} \frac{1}{2} \| y - \mathbf{K}_{\mathcal{G}\mathcal{G}_s}(\mathbf{K}_{\mathcal{G}_s\mathcal{G}_s} + \epsilon \mathbf{I}) \mathbf{Y}_s^{h^i} \|_F^2, \tag{2}$$

where $\mathbf{K}_{\mathcal{G}\mathcal{G}_s}: \mathcal{G} \times \mathcal{G}_s \rightarrow \mathbb{R}^{N \times m}$ denotes the kernel function, it is a simple algorithm that involving the structure to the kernel calculation. We adopt SNTK (Wang et al., 2024) for calculating the node-level and LightGNTK (Xu et al., 2023) for graph-level calculating. $\mathbf{Y}_s^{h^i}$ denotes the corresponding label sets in the hard condensed labels $\mathbf{Y}_s^h$. Then $\mathcal{G}_s$ could be adapted to $\mathcal{G}_s^{i'}$ by Eq. (1), which represents that the condensed graph $\mathcal{G}_s$ update through task $\mathcal{G}^{T_i}$. The $\| \cdot \|_F^2$ indicates the Mean Square Error (MSE) loss function.

### 3.3 CROSS-DATASET TRANSFERABILITY: MULTI-TEACHER SELF-SUPERVISED OPTIMIZATION

Regarding the "cross-dataset" transferability challenge, the natural approach is to extract and distill fundamental knowledge into the condensed graph. Beyond human-made labels (supervised task), we propose leveraging self-supervised tasks to automatically extract latent knowledge. Furthermore, as each task preserves only one aspect of the dataset (Jin et al., 2022a), we propose to use multi-teacher self-supervised optimization to better consider the diverse aspects of the dataset.

**Unifying Self-Supervised Task Labels.** Directly incorporating self-supervised tasks is not trivial, as existing graph condensation methods require explicit labels. For classification tasks with $C$ classes, synthetic labels $\mathbf{Y}_s \in \mathbb{R}^{m \times C}$ must be assigned to $\mathcal{G}_s$. Assigning numerous synthetic labels for multiple self-supervised tasks is impractical. In addition, simultaneously training such tasks from scratch in condensing, as in previous graph condensation methods, is complex and time-consuming.

Inspired by the knowledge distillation works (Fan et al., 2024), we opt to load the pre-trained models as teachers, and then we can unify all the tasks to the target $\left( \sum_{i=1}^{k} \lambda_i f_i^{\mathcal{T}}(\mathbf{X}, \mathbf{A}) \right) \in \mathbb{R}^{N \times d}$, where $\{\lambda_i\}_{i=1}^{k}$ are the weights of teachers and we discuss it in Sec. 3.4. Thus all the target pseudo labels turn to be same $d$-dimension tensors. We only need to craft a soft condensed label $\mathbf{Y}_s^s \in \mathbb{R}^{m \times d}$ in condensing:

$$\min_{\mathbf{Y}_s^s} \mathcal{L}_{self}(\mathcal{P}^{\mathcal{T}}(\mathbf{X}, \mathbf{A}), \mathbf{Y}_s^s), \tag{3}$$

where $\mathcal{P}^{\mathcal{T}}(\mathbf{X}, \mathbf{A}) = \left( \sum_{i=1}^{k} \lambda_i f_i^{\mathcal{T}}(\mathbf{X}, \mathbf{A}) \right)$ denotes the adjusted distribution of the pre-trained teacher models' output. It is also worth noting that loading the pre-trained tasks instead of training a model from scratch, as existing methods do, can save considerable optimization time and resource.

**Kernel Ridge Regression-Based Optimizing Objective.** Here we also use kernel ridge regression (KRR) (Lee et al., 2024) for optimizing the $\mathcal{G}_s$ and the soft condensed label $\mathbf{Y}_s^s$. The motivation is that the naive gradient matching and trajectory matching methods are hard to support continuous labels and are heavily coupled with supervised tasks. Formally, the KRR-based self-supervised loss $\mathcal{L}_{self}$ is:

$$\mathcal{L}_{self} \triangleq \min_{\mathcal{G}_s} \frac{1}{2} \| \mathcal{P}^{\mathcal{T}}(\mathbf{X}, \mathbf{A}) - \mathbf{K}_{\mathcal{G}\mathcal{G}_s}(\mathbf{K}_{\mathcal{G}_s\mathcal{G}_s} + \epsilon\mathbf{I})\mathbf{Y}_s^s \|_F^2, \tag{4}$$

where $\mathbf{K}(\cdot)$ is similar to that in Eq. (1), $\mathbf{I}$ is the identity matrix improving the robustness, and $\mathcal{P}^{\mathcal{T}}(\mathbf{X}, \mathbf{A})$ represents the weighted combination of the teacher models' output.

## 3.4 MUTUAL INFORMATION GUIDED JOINT CONDENSATION

Following the existing KRR-based graph condensation methods (Xu et al., 2023; Wang et al., 2024), the overall condensing objective is the weight sum of $\mathcal{L}_{cls}^{meta}$ and $\mathcal{L}_{self}$ as: $\min_{\mathcal{G}_s} \mathcal{L} = \mathcal{L}_{cls}^{meta} + \alpha\mathcal{L}_{self}$. However, we empirically find there may exist a potential conflict in calibrating multiple self-supervised and supervised tasks (see Table 6). Such findings drive us to find an optimal guiding solution in adjusting.

As similar issues discussed in the previous knowledge distilling literature (Fan et al., 2024; Wu et al., 2022), the key to condensing the beneficial information from teachers is to let the probability distribution of teachers approximate the true (Baysian) distribution of the downstream tasks (Menon et al., 2021). However, due to the invisibility of downstream labels, we intuitively use the label $\mathbf{Y}$ from $\mathcal{G}$ to serve as the target ideal distribution. The mutual information to calculate the dependency between the $\mathcal{P}^{\mathcal{T}}(\mathbf{X}, \mathbf{A})$ and labels:

$$I(\mathcal{P}^{\mathcal{T}}(\mathbf{X}, \mathbf{A}); \mathbf{Y}) = H(\mathbf{Y}) - H(\mathbf{Y}|\mathcal{P}^{\mathcal{T}}(\mathbf{X}, \mathbf{A})), \tag{5}$$

where $\mathcal{P}^{\mathcal{T}}(\mathbf{X}, \mathbf{A})$ represents a weighted combination of the multi-teacher output. However, directly solving Eq. (5) is computationally intensive and may not yield optimal results due to the sampling process involved in Sec. 3.2. To address this issue, we propose optimizing the mutual information $\mathcal{L}_{MI} = I(\mathbf{Y}_s^s; \mathbf{Y}_s^h)$ as a substitution (Yuan et al., 2024; Fu et al., 2024). We further demonstrate that the mutual information of the condensed graph labels, $I(\mathbf{Y}_s^s; \mathbf{Y}_s^h)$ is related to the upper bound of $I(\mathcal{P}^{\mathcal{T}}(\mathbf{X}, \mathbf{A}); \mathbf{Y})$:

**Theorem 1.** *Given that $\mathbf{Y_s^s}$ and $\mathbf{Y_s^h}$ are approximations through kernel ridge regression, the mutual information of the condensed graph $I(\mathbf{Y_s^s}; \mathbf{Y_s^h})$ is related to the upper bound of the mutual information $I(\mathcal{P}^{\mathcal{T}}(\mathbf{X}, \mathbf{A}); \mathbf{Y})$ as follows:*

$$I(\mathbf{Y_s^s}; \mathbf{Y_s^h}) \leq I(\mathcal{P}^{\mathcal{T}}(\mathbf{X}, \mathbf{A}); \mathbf{Y}). \tag{6}$$

The detailed proof is provided in Appendix C. In our experiments, we implement the neural estimation method (Belghazi et al., 2018) for the gradient backward pass. The final objective function is expressed as: $\min_{\mathcal{G}_s} \mathcal{L} = \mathcal{L}_{cls}^{meta} + \alpha\mathcal{L}_{self} + \beta\mathcal{L}_{MI}$. The overall algorithm is presented in Appendix F.1.

## 4 EXPERIMENT

### 4.1 EXPERIMENTAL SETUPS

**Datasets.** We evaluate our method on 6 node-level datasets (Cora (Kipf & Welling, 2017), Citeseer (Kipf & Welling, 2017), Ogbn-arxiv (Hu et al., 2020a), Reddit (Hamilton et al., 2017) and Flickr (Zeng et al., 2020)) and 5 graph-level datasets (GEOM (Axelrod & Gomez-Bombarelli, 2020), BACE (Wu et al., 2018), ClinTox (Gayvert et al., 2016), and SIDER (Kuhn et al., 2016)). For the supervised node classification task, we follow the settings from GCond (Jin et al., 2022c). For the other types of task, we follow the public split of the dataset. We report the details of the datasets in Appendix A.

**Baselines.** We compare `ST-GCond` with 10 baselines: (1) graph coreset methods (`Random`, `Herding` (Welling, 2009), and `K-Center` (Wolf, 2011)), (2) node-level graph condensation methods (`GCond` (Jin et al., 2022c), `SGDD` (Yang et al., 2023), `SFGC` (Zheng et al., 2023), and `GEOM` (Zhang et al., 2024)), (3) graph-level graph condensation methods (`DosCond` (Jin et al., 2022b), `KiDD` (Xu et al., 2023)), and (4) one self-supervised condensation method, `KRR-ST` (Lee et al., 2024). For the `Random` method, we randomly select nodes from the original graph to induce a subgraph. We implement `Herding` to filter nodes that are close to the centroids and `K-Center` to select nodes that play a critical role in minimizing the distance between samples and their nearest centers following (Jin et al., 2022c).

**Implementation Details.** *In the condensing stage,* for the supervised task, we randomly split the classes of the task into 3 parts, with each part containing $h$ classes ($h > 1$) as a sub-task. For the auxiliary self-supervised tasks, we select 5 classic tasks for node-level condensation (DGI (Velickovic et al., 2019), CLU (You et al., 2020b), PAR (You et al., 2020b), PAIRSIM (Jin et al., 2020), and PAIRDIS (Peng et al., 2020)) and 7 tasks for graph-level condensation ( AttrMask (Hu et al., 2020b), ContextPred (Hu et al., 2020b), EdgePred (Hamilton et al., 2017), GPT-GNN (Hu et al., 2020c), GraphLoG (Xu et al., 2021), GraphCL (You et al., 2020a), and InfoGraph(Sun et al., 2019)). We briefly introduce them in Appendix B. For each dataset, the condensation ratio $r$ is defined by the number of nodes in the condensed graph $r = \frac{N}{m}$, where $0 < r < 1$. For each dataset, we condense 5 graphs with different random seeds and report average performance. We provide our

*In the testing stage,* we choose the appropriate testing paradigm according to the specific scenario. (1) *Single dataset and task scenario.* When the downstream dataset and task match the condensing one, similar to the previous graph dataset condensation methods, we use $\{\mathbf{X_s}, \mathbf{A_s}, \mathbf{Y_s^h}, \mathbf{Y_s^s}\}$ or $\{\mathbf{X_s}, \mathbf{A_s}, \mathbf{Y_s^h}\}$ to train a model and test it on the original graph dataset $\mathcal{G}$. (2) *cross-dataset and cross-task scenario.* When the downstream dataset and task differ, we use $\{\mathbf{X_s}, \mathbf{A_s}, \mathbf{Y_s^s}\}$ to train a model and then use the downstream data to train a linear classifier. For other graph condensation methods, we replace the last layer of the trained model with a linear classifier in a similar manner. Our code is available at https://github.com/RingBDStack/ST-GCond.

## 4.2 EXPERIMENTAL RESULTS

To evaluate the performance of `ST-GCond`, we conduct experiments on three scenarios: (1) single task and dataset scenario; (2) cross-dataset scenario; (3) cross-task scenario.

**(1) Performance comparison on single dataset and task.** In this scenario, `ST-GCond` utilizes $\mathbf{Y_s^s}$ to train a GNN feature extractor and further uses $\mathbf{Y_s^h}$ to train a linear classifier. We also report a variant `ST-GCond`-$\mathbf{Y_s^h}$ of `ST-GCond`, which exclusively uses $\mathbf{Y_s^h}$ during the training stage. As shown in Table 1, `ST-GCond` achieves state-of-the-art results in 14 out of 15 ratios across 5 datasets, particularly excelling at the lowest ratio. The maximum improvement is 2.71%. Furthermore, `ST-GCond` outperforms `ST-GCond`-$\mathbf{Y_s^h}$ variant by an average margin of 0.5% to 1.9%.

From the experiment, we observe that while our method is specifically designed for cross-dataset and cross-task scenarios, it also demonstrates non-trivial improvements in single-dataset and single-task settings. Notably, in three out of five datasets, the results are even lossless compared to whole dataset accuracy according to the definition from `GEOM` (Zhang et al., 2024). We attribute this improvement to the integration of both supervised and self-supervised information. This approach effectively mimics the property in the original dataset, as highlighted in various studies (Wang et al., 2019; Hou et al., 2022; Liu et al., 2023b), which emphasize the importance of self-supervised learning in enhancing performance. Furthermore, the `ST-GCond`-$\mathbf{Y_s^h}$ also achieves the comparable performance comparing to the baselines, demonstrating the versatility of using the hard labels $\mathbf{Y_h^s}$ independently with the condensed graph.

**(2) Performance comparison on cross-dataset scenario.** We compare `ST-GCond` with baselines under cross-dataset transfer learning settings. For node-level datasets, we use `Ogbn-arxiv` as the source dataset and test the condensed graph on four other target datasets. For graph-level datasets, we use `GEOM` as the source dataset and test the condensed graph on four other target datasets. We compare `ST-GCond` with `GCond`, `SGDD`, `SFGC`, and `GEOM` for node-level condensation, and with `DosCond` and `KiDD` for graph-level condensation. `MLP (w/o pre)` shows the naive results that solely use the linear model in the target datasets, a common strategy for low-resource computation. The results are shown in Table 2 and Table 3.

Table 1: Node classification performance (Accuracy%± std) comparison under the **single task and dataset scenario**. Best results are in **bold** and the runner-up is underlined. We use red text to indicate the lossless results (better than solely using GCN on the original graph).

| Datasets | Ratio($r$) | Random $(\mathbf{A_s,X_s,Y_s})$ | Herding $(\mathbf{A_s,X_s,Y_s})$ | K-Center $(\mathbf{A_s,X_s,Y_s})$ | GCond $(\mathbf{A_s,X_s,Y_s})$ | SGDD $(\mathbf{A_s,X_s,Y_s})$ | SFGC $(\mathbf{X_s,Y_s})$ | GEOM $(\mathbf{X_s,Y_s})$ | ST-GCond-$Y_s^h$ $(\mathbf{A,X,Y_s^h})$ | ST-GCond $(\mathbf{A,X,Y_s^a,Y_s^h})$ | Whole Dataset |
|---|---|---|---|---|---|---|---|---|---|---|---|
| Citeseer | 0.90% | $54.4_{\pm4.4}$ | $57.1_{\pm1.5}$ | $52.4_{\pm2.8}$ | $70.5_{\pm1.2}$ | $69.5_{\pm0.4}$ | $70.4_{\pm0.1}$ | $69.8_{\pm0.5}$ | $\underline{71.5_{\pm0.8}}$ | $\mathbf{71.5_{\pm0.5}}$ | $71.7_{\pm0.1}$ |
| | 1.80% | $64.2_{\pm1.7}$ | $66.7_{\pm1.0}$ | $64.3_{\pm1.0}$ | $70.6_{\pm0.9}$ | $70.2_{\pm0.8}$ | $70.1_{\pm0.3}$ | $\underline{70.8_{\pm0.7}}$ | $69.8_{\pm1.1}$ | $\mathbf{71.6_{\pm0.7}}$ | |
| | 3.60% | $69.1_{\pm0.1}$ | $69.0_{\pm0.1}$ | $69.1_{\pm0.1}$ | $69.8_{\pm1.4}$ | $70.3_{\pm1.7}$ | $\underline{71.4_{\pm0.8}}$ | $70.2_{\pm0.3}$ | $69.6_{\pm0.9}$ | $\mathbf{72.1_{\pm0.3}}$ | |
| Cora | 1.30% | $63.6_{\pm3.7}$ | $67.0_{\pm1.3}$ | $64.0_{\pm2.3}$ | $79.8_{\pm1.3}$ | $80.1_{\pm0.7}$ | $80.1_{\pm0.4}$ | $82.5_{\pm0.4}$ | $\underline{81.8_{\pm0.9}}$ | $\mathbf{83.4_{\pm0.8}}$ | $81.2_{\pm0.2}$ |
| | 2.60% | $72.8_{\pm1.1}$ | $73.4_{\pm1.0}$ | $73.2_{\pm1.2}$ | $80.1_{\pm0.6}$ | $80.6_{\pm0.8}$ | $81.7_{\pm0.5}$ | $\mathbf{83.6_{\pm0.3}}$ | $81.4_{\pm1.0}$ | $\underline{83.3_{\pm0.5}}$ | |
| | 5.20% | $76.8_{\pm0.1}$ | $76.8_{\pm0.1}$ | $76.6_{\pm0.1}$ | $79.3_{\pm0.3}$ | $80.4_{\pm1.6}$ | $81.6_{\pm0.8}$ | $82.8_{\pm0.7}$ | $81.8_{\pm0.8}$ | $\mathbf{83.6_{\pm0.9}}$ | |
| Ogbn-arxiv | 0.05% | $47.1_{\pm3.9}$ | $52.4_{\pm1.8}$ | $47.2_{\pm3.0}$ | $59.2_{\pm1.1}$ | $60.8_{\pm1.3}$ | $65.5_{\pm0.7}$ | $\underline{65.5_{\pm0.4}}$ | $65.1_{\pm1.1}$ | $\mathbf{66.8_{\pm0.8}}$ | $71.4_{\pm0.1}$ |
| | 0.25% | $57.3_{\pm1.1}$ | $58.6_{\pm1.2}$ | $56.8_{\pm0.8}$ | $63.2_{\pm0.3}$ | $65.8_{\pm1.2}$ | $66.1_{\pm0.4}$ | $\underline{65.6_{\pm0.2}}$ | $65.6_{\pm1.2}$ | $\mathbf{66.8_{\pm0.9}}$ | |
| | 0.50% | $60.0_{\pm0.9}$ | $60.4_{\pm0.8}$ | $60.3_{\pm0.4}$ | $64.0_{\pm0.4}$ | $66.3_{\pm0.7}$ | $66.8_{\pm0.4}$ | $67.6_{\pm0.3}$ | $\underline{68.5_{\pm0.8}}$ | $\mathbf{68.1_{\pm0.3}}$ | |
| Flickr | 0.10% | $41.8_{\pm2.0}$ | $42.5_{\pm1.8}$ | $42.0_{\pm0.7}$ | $46.5_{\pm0.3}$ | $46.9_{\pm0.3}$ | $46.6_{\pm0.2}$ | $\underline{47.1_{\pm0.1}}$ | $46.8_{\pm0.1}$ | $\mathbf{47.2_{\pm0.1}}$ | $47.2_{\pm0.1}$ |
| | 0.50% | $44.0_{\pm0.4}$ | $43.9_{\pm0.9}$ | $43.2_{\pm0.1}$ | $47.1_{\pm0.1}$ | $47.1_{\pm0.3}$ | $47.0_{\pm0.2}$ | $47.0_{\pm0.2}$ | $47.0_{\pm0.2}$ | $\mathbf{47.5_{\pm0.3}}$ | |
| | 1.00% | $44.6_{\pm0.2}$ | $44.4_{\pm0.6}$ | $44.1_{\pm0.4}$ | $47.1_{\pm0.4}$ | $47.1_{\pm0.1}$ | $47.1_{\pm0.1}$ | $\underline{47.3_{\pm0.3}}$ | $47.1_{\pm0.3}$ | $\mathbf{47.5_{\pm0.4}}$ | |
| Reddit | 0.05% | $46.1_{\pm4.4}$ | $53.1_{\pm2.5}$ | $46.6_{\pm2.3}$ | $88.0_{\pm1.8}$ | $90.5_{\pm2.1}$ | $89.7_{\pm0.2}$ | $91.1_{\pm0.4}$ | $\underline{91.4_{\pm0.4}}$ | $\mathbf{91.8_{\pm0.4}}$ | $93.9_{\pm0.0}$ |
| | 0.10% | $58.0_{\pm2.2}$ | $62.7_{\pm1.0}$ | $53.0_{\pm3.3}$ | $89.6_{\pm0.7}$ | $91.8_{\pm1.9}$ | $90.0_{\pm0.3}$ | $\underline{91.4_{\pm0.2}}$ | $91.5_{\pm0.2}$ | $\mathbf{91.7_{\pm0.2}}$ | |
| | 0.20% | $66.3_{\pm1.9}$ | $71.0_{\pm1.6}$ | $58.5_{\pm2.1}$ | $90.1_{\pm0.5}$ | $91.6_{\pm1.8}$ | $89.9_{\pm0.4}$ | $\underline{91.5_{\pm0.4}}$ | $91.9_{\pm0.4}$ | $\mathbf{92.4_{\pm0.4}}$ | |

Table 2: Node classification performance (Accuracy% ± std) comparison under the **cross-dataset scenario**. Best results are in **bold** and the runner-up is underlined. We use the red text to indicate the lossless results (better than solely using GCN on the target datasets).

| Methods | \multicolumn Node-level: Ogbn-arxiv → Target datasets | | | | | | | | | | | |
|---|---|---|---|---|---|---|---|---|---|---|---|---|
| | Cora | | | Citeseer | | | Flickr | | | Reddit | | |
| | 0.05% | 0.25% | 0.50% | 0.05% | 0.25% | 0.50% | 0.05% | 0.25% | 0.50% | 0.05% | 0.25% | 0.50% |
| MLP (w/o pre) | $54.8_{\pm0.8}$ | | | $60.1_{\pm1.7}$ | | | $28.75_{\pm1.1}$ | | | $68.5_{\pm0.8}$ | | |
| Random | $42.9_{\pm2.8}$ | $41.9_{\pm1.2}$ | $43.7_{\pm1.8}$ | $58.0_{\pm2.8}$ | $59.1_{\pm1.7}$ | $58.1_{\pm1.9}$ | $26.0_{\pm2.0}$ | $27.3_{\pm2.4}$ | $28.1_{\pm1.7}$ | $68.2_{\pm2.1}$ | $68.3_{\pm1.4}$ | $69.0_{\pm0.8}$ |
| Herding | $48.7_{\pm1.9}$ | $47.3_{\pm2.5}$ | $50.2_{\pm3.0}$ | $62.5_{\pm3.3}$ | $64.1_{\pm2.8}$ | $66.8_{\pm3.5}$ | $28.0_{\pm2.0}$ | $29.1_{\pm2.5}$ | $29.7_{\pm2.8}$ | $73.8_{\pm2.2}$ | $74.7_{\pm2.4}$ | $75.3_{\pm2.6}$ |
| GCond | $\underline{65.3_{\pm1.6}}$ | $63.7_{\pm2.2}$ | $\underline{69.7_{\pm2.3}}$ | $\underline{67.3_{\pm1.8}}$ | $61.9_{\pm1.8}$ | $\underline{64.6_{\pm5.4}}$ | $33.2_{\pm1.3}$ | $29.9_{\pm1.6}$ | $29.7_{\pm0.5}$ | $\underline{86.2_{\pm1.3}}$ | $84.9_{\pm0.4}$ | $82.8_{\pm2.1}$ |
| SGDD | $65.1_{\pm2.0}$ | $63.6_{\pm2.0}$ | $67.0_{\pm2.2}$ | $70.1_{\pm2.0}$ | $66.4_{\pm5.0}$ | $67.8_{\pm5.0}$ | $35.2_{\pm1.3}$ | $33.5_{\pm2.7}$ | $32.4_{\pm2.3}$ | $85.8_{\pm1.1}$ | $85.2_{\pm1.2}$ | $84.1_{\pm1.8}$ |
| SFGC | $65.0_{\pm2.3}$ | $63.5_{\pm0.5}$ | $64.3_{\pm2.4}$ | $62.9_{\pm2.1}$ | $60.9_{\pm9.5}$ | $61.0_{\pm5.8}$ | $\underline{37.3_{\pm0.9}}$ | $\underline{37.2_{\pm0.4}}$ | $\underline{35.2_{\pm1.6}}$ | $85.4_{\pm0.9}$ | $85.5_{\pm0.8}$ | $85.4_{\pm2.4}$ |
| GEOM | $61.5_{\pm0.9}$ | $\underline{66.0_{\pm1.1}}$ | $65.9_{\pm2.1}$ | $64.0_{\pm2.6}$ | $\underline{68.9_{\pm0.8}}$ | $67.7_{\pm2.4}$ | $36.2_{\pm0.6}$ | $34.2_{\pm1.7}$ | $34.7_{\pm0.7}$ | $84.5_{\pm1.2}$ | $\underline{87.1_{\pm1.0}}$ | $\underline{87.5_{\pm0.8}}$ |
| ST-GCond | $\mathbf{74.1_{\pm0.8}}$ | $\mathbf{81.5_{\pm1.1}}$ | $\mathbf{81.8_{\pm1.8}}$ | $\mathbf{69.8_{\pm1.7}}$ | $\mathbf{71.4_{\pm0.7}}$ | $\mathbf{72.8_{\pm0.6}}$ | $\mathbf{43.6_{\pm0.6}}$ | $\mathbf{47.2_{\pm0.1}}$ | $\mathbf{47.8_{\pm0.4}}$ | $\mathbf{88.7_{\pm0.4}}$ | $\mathbf{90.8_{\pm0.9}}$ | $\mathbf{92.1_{\pm0.8}}$ |

We observe that ST-GCond exceeds all runner-up methods by an average of 2.5% to 15.5% on node-level classification and 4.1% to 18.79% on graph classification. These improvements demonstrate the effectiveness of incorporating self-supervised tasks to extract "universal knowledge," enabling the condensed graph to benefit various downstream datasets. While corset methods perform similarly to MLP results, existing graph condensation methods show improvements over MLP, indicating the versatility of the condensed graph. Additionally, ST-GCond achieves better results on target datasets than using the GCN model alone. Thus, downstream users can achieve similar test performance to expensive GCN models with significantly lower computational costs by training models on the condensed graph and using a simple linear classifier.

**(3) Performance comparison on cross-task scenario.** In the cross-task scenario, our definition of "task" encompasses two distinct meanings. First, it refers to different task types, such as node classification, node clustering, and link prediction, which we term the "type-changing cross-task setting." Second, it refers to the variation in class labels within the same type of supervised task, which we define as the "label-set changing cross-task setting." Such two settings are widely We conduct experiments in both of these settings to validate our approach.

In the task type changing cross-task setting, we use node classification for condensation and link prediction or node clustering as downstream tasks. The results are shown in Table 4 and Table 5. We use VGAE (Kipf & Welling, 2016) and ARGA (Pan et al., 2018) as baselines. As shown in Table 4, while existing methods produce comparable results, they are sub-optimal compared to the baselines. In contrast, ST-GCond achieves better AUC and AP metrics on Cora, and the AP metric on Citeseer, indicating that our condensed graph retains beneficial knowledge for cross-task scenarios. Table 5 shows more pronounced improvements among existing methods, which can be attributed to the inherent relationship between clustering and labels (Jin et al., 2020).

In the label-set changing cross-task setting, the dataset's label set is divided into two parts: one for condensation and the other for downstream tasks. Figure 3 shows the performance over test

Table 3: Graph classification performance (ROC-AUC(%) ± std) comparison under the **cross-dataset scenario**. Best results are in **bold** and the runner-up is underlined. We use the red text to indicate the lossless results (better than solely using GCN on the original graph)

| Methods | Graph-level: GEOM-data → Target datasets | | | | | | | | | | | |
| | BACE | | | BBBP | | | ClinTox | | | SIDER | | |
| | 0.001% | 0.005% | 0.01% | 0.001% | 0.005% | 0.01% | 0.001% | 0.005% | 0.01% | 0.001% | 0.005% | 0.01% |
| MLP (w/o pre) | $48.7_{\pm0.3}$ | | | $49.3_{\pm0.8}$ | | | $43.7_{\pm0.3}$ | | | $47.1_{\pm1.6}$ | | |
| Random | $43.6_{\pm0.4}$ | $45.8_{\pm0.7}$ | $46.2_{\pm0.7}$ | $38.5_{\pm0.3}$ | $33.1_{\pm0.3}$ | $36.2_{\pm0.8}$ | $42.8_{\pm0.8}$ | $43.5_{\pm1.2}$ | $45.8_{\pm0.3}$ | $43.7_{\pm1.2}$ | $48.5_{\pm2.8}$ | $44.2_{\pm1.2}$ |
| Herding | $47.0_{\pm0.5}$ | $49.0_{\pm0.7}$ | $50.0_{\pm0.6}$ | $40.0_{\pm0.4}$ | $38.0_{\pm0.5}$ | $39.5_{\pm0.6}$ | $45.0_{\pm0.6}$ | $46.5_{\pm0.8}$ | $47.0_{\pm0.5}$ | $46.0_{\pm1.0}$ | $47.5_{\pm1.5}$ | $48.0_{\pm1.3}$ |
| DosCond | $54.6_{\pm0.8}$ | $55.7_{\pm0.1}$ | $51.7_{\pm0.3}$ | $47.0_{\pm0.7}$ | $48.5_{\pm0.9}$ | $49.0_{\pm0.8}$ | $50.5_{\pm0.9}$ | $51.0_{\pm1.0}$ | $52.0_{\pm0.7}$ | $50.5_{\pm1.2}$ | $52.0_{\pm1.5}$ | $53.0_{\pm1.3}$ |
| KiDD | $53.8_{\pm0.7}$ | $53.5_{\pm0.8}$ | $54.8_{\pm1.7}$ | $52.8_{\pm0.9}$ | $54.0_{\pm0.8}$ | $55.0_{\pm0.7}$ | $55.0_{\pm0.8}$ | $56.0_{\pm0.9}$ | $57.0_{\pm0.6}$ | $51.5_{\pm0.8}$ | $51.1_{\pm1.0}$ | $51.0_{\pm0.7}$ |
| ST-GCond | $\mathbf{68.6}_{\pm1.0}$ | $\mathbf{71.4}_{\pm0.8}$ | $\mathbf{73.6}_{\pm1.1}$ | $\mathbf{58.6}_{\pm1.1}$ | $\mathbf{61.4}_{\pm0.8}$ | $\mathbf{62.8}_{\pm0.7}$ | $\mathbf{64.1}_{\pm0.7}$ | $\mathbf{64.8}_{\pm0.6}$ | $\mathbf{71.5}_{\pm0.4}$ | $\mathbf{55.6}_{\pm0.3}$ | $\mathbf{56.8}_{\pm0.8}$ | $\mathbf{55.7}_{\pm0.3}$ |

Table 4: Link prediction results under the **cross-task scenarios**, where each graph condensation method observes only the classification task information.

| Method | Cora | | Citeseer | |
| | AUC(%) | AP(%) | AUC(%) | AP(%) |
| VGAE | 91.4 | 92.6 | 90.8 | 92.0 |
| ARGA | 92.4 | 93.2 | 91.9 | 92.1 |
| GCond | 81.6↓ | 83.7↓ | 73.5↓ | 74.8↓ |
| SGDD | 85.4↓ | 88.3↓ | 74.3↓ | 76.9↓ |
| SFGC | 83.3↓ | 85.1↓ | 73.6↓ | 73.9↓ |
| GEOM | 84.7↓ | 85.0↓ | 73.2↓ | 74.8↓ |
| ST-GCond | **93.3**↑ | **94.8**↑ | 90.6↓ | **92.3**↑ |

Table 5: Node clustering results under the **cross-task scenarios**. We use the **bold** denotes the best results, the underline indicates the runner-ups. ↑/↓ indicate the increase/decrease compared to baseline VGAE/ARGA.

| Method | Cora | | | Citeseer | | |
| | NMI(%) | F1(%) | ARI(%) | NMI(%) | F1(%) | ARI(%) |
| VGAE | 51.4 | 57.5 | 38.7 | 34.8 | 55.4 | 28.5 |
| ARGA | 50.8 | 65.6 | 34.7 | 40.0 | 54.6 | 34.1 |
| GCond | 48.6↓ | 54.4↓ | 36.8↓ | 31.6↓ | 52.8↓ | 27.4↓ |
| SGDD | 51.8↑ | 69.4↑ | 38.7↑ | 35.1↓ | 58.8↑ | 26.5↓ |
| SFGC | 50.7↓ | 68.2↑ | 36.9↓ | 34.8↓ | 58.1↑ | 25.6↓ |
| GEOM | 48.2↓ | 55.7↓ | 32.9↓ | 33.1↓ | 49.3↓ | 21.3↓ |
| ST-GCond | **53.8**↑ | **71.4**↑ | **40.7**↑ | **40.8**↑ | **62.7**↑ | **41.7**↑ |

epochs. Since the model trained on the condensed graph cannot be directly applied to the downstream task, the final layer is replaced with a linear layer and fine-tuned with the target training data. Our task-disentangled meta-optimization strategy allows the condensed graph to acquire cross-task knowledge, evidenced by ST-GCond achieving faster and superior results compared to other methods, particularly the naive GCN (gray line), demonstrating the versatility of our approach.

## 4.3 ABLATION STUDY AND SENSITIVITY ANALYSIS

**Ablation study.** To investigate the impact of task-disentangled meta optimization, multi-teacher self-supervised optimization, and MI-guided joint condensation, we conduct an ablation study presented in Table 6. We select cross-dataset experiments to evaluate the transferability of these components. We take Ogbn-arxiv as the source dataset and evaluate on Cora and Reddit. The ablations include: *w/o self* (excluding self-supervised tasks), *w/o meta* (excluding task disentanglement), and *w/o MI* (excluding mutual information constraints). $r$ denotes the condensation ratio for Ogbn-arxiv.

Table 6: Mean accuracy(%) and standard deviation of node classification.

| Methods | Ogbn-arxiv → Cora (Ratio $r$) | | Ogbn-arxiv → Reddit (Ratio $r$) | |
| | 0.25% | 0.50% | 0.25% | 0.50% |
| Random | $59.1_{\pm1.7}$ | $58.1_{\pm1.9}$ | $68.3_{\pm1.4}$ | $69.0_{\pm0.8}$ |
| Herding | $47.3_{\pm2.5}$ | $50.2_{\pm3.0}$ | $74.7_{\pm2.4}$ | $75.3_{\pm2.6}$ |
| SGDD | $63.6_{\pm2.0}$ | $67.0_{\pm2.2}$ | $85.2_{\pm1.2}$ | $84.1_{\pm1.8}$ |
| GEOM | $66.0_{\pm1.1}$ | $65.9_{\pm2.1}$ | $87.1_{\pm1.0}$ | $87.5_{\pm0.8}$ |
| ST-GCond w/o self | $78.4_{\pm0.9}$ | $77.4_{\pm1.0}$ | $88.6_{\pm0.9}$ | $89.0_{\pm1.2}$ |
| ST-GCond w/o meta | $76.4_{\pm0.2}$ | $73.4_{\pm0.7}$ | $88.9_{\pm0.8}$ | $89.7_{\pm0.4}$ |
| ST-GCond w/o MI | $68.5_{\pm0.7}$ | $74.4_{\pm1.1}$ | $87.4_{\pm1.1}$ | $87.4_{\pm0.8}$ |
| ST-GCond | $\mathbf{81.5}_{\pm1.1}$ | $\mathbf{81.8}_{\pm1.8}$ | $\mathbf{90.8}_{\pm0.9}$ | $\mathbf{92.1}_{\pm0.8}$ |

As observed, *ST-GCond w/o self* and *ST-GCond w/o meta* demonstrate strong performance relative to the baselines, likely due to the shortcomings of naive methods, which tend to overemphasize task-specific information. However, both variants still fall short when compared to the results obtained using the full dataset (e.g., accuracies of 81.2% on Cora and 93.9% on Reddit), emphasizing the necessity of combining these components for optimal performance. Nevertheless, the naive combination of such two parts (*ST-GCond w/o MI*) underperforms both individual variants, . Sim-

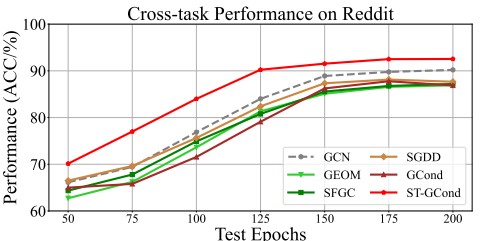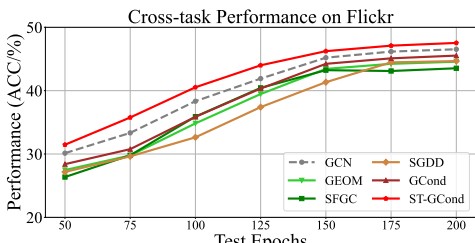

Figure 3: Displaying performance v.s. test epochs under the **cross-task scenarios**. Except for the GCN curve (the gray line), all other methods only observe the other half of the supervised information.

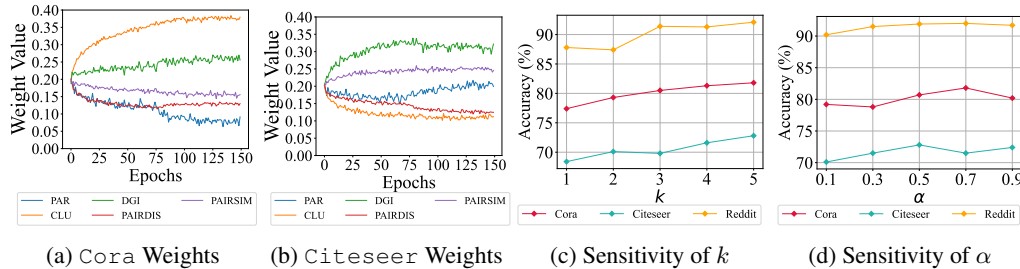

(a) `Cora` Weights    (b) `Citeseer` Weights    (c) Sensitivity of $k$    (d) Sensitivity of $\alpha$

Figure 4: Ablation Study and Sensitivity Analysis. Fig. (a) and Fig. (b) illustrate the convergence of adaptive task weights by the end of training. Fig. (c) shows an increasing performance trend with respect to the number of teachers ($k$), while Fig. (d) does not show a significant trend.

ilar observations have been discussed in prior works Jin et al. (2022a); Fan et al. (2024), further highlighting the critical role of $\mathcal{L}_{MI}$ in resolving such conflicts and improving overall functionality.

**Evolution process of teachers' weights** $\{\lambda_i\}_{i=1}^k$**.** As discussed in Sec. 3.4, the combination of the teachers' outputs should approximate the true (Bayesian) distribution of the downstream tasks. However, since we cannot access the true distribution of the downstream tasks, we use $\mathbf{Y_s^h}$ as a surrogate target. Fig. 4a and 4b illustrate the evolution of weight values during the condensing process. It is observed that the same method may have different weights across various datasets, and the weights of the five tasks eventually stabilize and converge to steady values. According to Table 6, incorporating the weights of self-supervised tasks results in improvements ranging from 3.4% to 13%, highlighting the necessity of addressing conflicts among tasks.

**Sensitivity analysis of hyperparameters** $k$**,** $\alpha$**, and** $\beta$**.** We present a sensitivity analysis for two hyperparameters: the number of self-supervised tasks ($k$) and the loss weight ($\alpha$, $\beta$). Fig. 4c and 4d show that a larger $k$ generally results in better performance, indicating the effectiveness of the mutual information constraint strategy. For the hyperparameter $\alpha$ and $\beta$ (Appendix. G.1), the performance varies with different $\alpha$ and $\beta$ values, suggesting that they should be selected through a grid search.

## 5 CONCLUSION

We propose `ST-GCond`, a novel framework for self-supervised and transferable graph dataset condensation. Unlike existing works focus on condensation for a single task and dataset, our approach creates condensed dataset with higher data-level transferability, enhancing downstream models ability when it applied to various new datasets and tasks. We introduce task-disentangled meta-updating for cross-task knowledge preservation and incorporate multiple supervised tasks to extract "universal knowledge". Finally, to avoid the potential conflict of jointly using self-supervised and supervised information as the optimization directions, we leverage the mutual information loss term to guide the condensation process. Experiments demonstrate `ST-GCond`'s effectiveness in both single-task/single-dataset and cross-task/cross-dataset scenarios. **Limitations and Future Work:** While the condense ratio is adjustable, dynamic user demands require re-condensation each time, bringing potential computation costs.

ACKNOWLEDGMENT

The corresponding author is Qingyun Sun. The authors of this paper are supported by the National Natural Science Foundation of China through grants No.62225202 and No.62302023, and the Fundamental Research Funds for the Central Universities.

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

# Appendix

## A  DATASET DETAILS

We use 5 node-level graph datasets (`Cora`, `Citeseer` (Kipf & Welling, 2017), `Ogbn-arixv` (Hu et al., 2020a), `Flickr` (Zeng et al., 2020), and `Reddit` (Hamilton et al., 2017)) and 5 graph-level graph datasets (`GEOM` (Axelrod & Gomez-Bombarelli, 2020), `BACE` (Wu et al., 2018), `BBBP` (Martins et al., 2012), `ClinTox` (Gayvert et al., 2016), and `SIDER` (Kuhn et al., 2016)). We further provide the statics of datasets in Table A1.

Table A1: Statistics of datasets.

| Level | Dataset | # Classes / # Tasks | #Nodes / #Graphs | # Edges | # Features |
|-------|---------|---------------------|------------------|---------|-----------|
| Node-level | Cora | 7 | 2,708 | 5,429 | 1,433 |
| | Citeseer | 6 | 3,327 | 4,732 | 3,703 |
| | Ogbn-Arxiv | 40 | 169,343 | 1,166,243 | 128 |
| | Flickr | 7 | 89,250 | 899,756 | 500 |
| | Reddit | 210 | 232,965 | 57,307,946 | 602 |
| Graph-level | BACE | 1 | 1,513 | - | - |
| | BBBP | 1 | 2,039 | - | - |
| | ClinTox | 2 | 1,478 | - | - |
| | Sider | 27 | 1,427 | - | - |

## B  SELF-SUPERVISED TEACHER TASKS

In the paper, we utilize different self-supervised task guided models as the way we extract the "universal knowledge" from the original dataset $\mathcal{G}$. Here, we present each model we used.

For node-level classification tasks, we follow AutoSSL (Jin et al., 2022a) and adopt five classic tasks:

- **DGI** (Velickovic et al., 2019): Maximizes the different views' representations (graph v.s. nodes).
- **CLU** (You et al., 2020b): Predicts pseudo-labels from $K$-means clustering on node features.
- **PAR** (You et al., 2020b): Predicts pseudo-labels from Metis graph partition (Karypis & Kumar, 1998).
- **PAIRSIM** (Jin et al., 2020): Predicts pairwise feature similarity between nodes.
- **PAIRDIS** (Peng et al., 2020): Predicts the shortest path length between nodes.

In the graph-level classification tasks, we follow WAS (Fan et al., 2024) to adopt 7 classic tasks:

- **AttrMask** (Hu et al., 2020b): Learns the regularities of node/edge attributes.
- **ContextPred** (Hu et al., 2020b): Explores graph structures by predicting the contexts.
- **EdgePred** (Hamilton et al., 2017): Predicts the connectivity of node pairs.
- **GPT-GNN** (Hu et al., 2020c): Introduces an attributed graph generation task to pre-train GNNs.
- **GraphLoG** (Xu et al., 2021): Introduces a hierarchical prototype to capture the global semantic clusters.
- **GraphCL** (You et al., 2020a): Constructs specific contrastive views of graph data.
- **InfoGraph** (Sun et al., 2019): Maximizes the mutual information between the representations of the graph and substructures.

## C  PROOF OF THEOREM 1

*Proof.* We aim to prove that:

$$I(\mathbf{Y}_s^s; \mathbf{Y}_s^h) \leq I(\mathcal{P}^{\mathcal{T}}(\mathbf{X}, \mathbf{A}); \mathbf{Y}). \tag{A.1}$$

Since $\mathbf{Y}_s^s$ and $\mathbf{Y}_s^h$ are obtained from $\mathcal{P}^{\mathcal{T}}(\mathbf{X}, \mathbf{A})$ and $\mathbf{Y}$ through kernel ridge regression—which is a deterministic mapping, they can be expressed as:

$$\mathbf{Y}_s^s = f\left(\mathcal{P}^{\mathcal{T}}(\mathbf{X}, \mathbf{A})\right), \tag{A.2}$$

$$\mathbf{Y}_s^h = g(\mathbf{Y}), \tag{A.3}$$

where $f$ and $g$ are the regression functions.

According to the data processing inequality (Beaudry & Renner, 2012), applying deterministic functions to random variables does not increase mutual information. Therefore, we have:

$$I\left(\mathbf{Y}_s^s; \mathbf{Y}_s^h\right) \leq I\left(\mathcal{P}^{\mathcal{T}}(\mathbf{X}, \mathbf{A}); \mathbf{Y}\right). \tag{A.4}$$

$\square$

## D  TIME COMPLEXITY ANALYSIS

`ST-GCond` primarily consists of two parts: sampling sub-tasks for meta updating and involving self-supervised tasks to guide condensing. For the former, we can treat them as a composition of bi-level optimization. Following `GCond` (Jin et al., 2022c), we start with an $L$-layer GCN, where the large-scale graph has $N$ nodes, the small yet informative condensed graph has $m$ nodes, and the hidden dimension is $d$. The computation cost for a single task involves a forward pass through the GNN, which is $O(Lm^2d + Lmd)$, and through $g_\phi$, which is $O(m^2d^2)$. The inner optimization of kernel ridge regression can be expressed as $O(Nmr^2 + Nm)$ (Wang et al., 2024). Therefore, the single task complexity is $O(Lm^2d + Lmd + m^2d^2 + Nmr^2 + Nm)$. Denoting the split of tasks as $t$, the complexity for the former part can be shown as $tO(Lm^2d + Lmd + m^2d^2 + Nmr^2 + Nm)$.

For the latter, the calculation process is similar to the former, although we introduce multiple self-supervised models during the condensing stage. Thanks to the benefits of the offline strategy, we only need the extra computation complexity of $kO(LEd + LNd^2)$, where $k$ denotes the number of self-supervised tasks. Therefore, the overall complexity can be expressed as $(t + 1)O(Lm^2d + Lmd + m^2d^2 + Nmr^2 + Nm) + kO(LEd + LNd^2)$. Note that $t$ and $k$ are not set to be too large.

To intuitively demonstrate the efficiency comparison, we present the running time (in seconds) of the proposed `ST-GCond` and `GCond` over 50 epochs on a single A100 GPU. Thanks to the efficiency of kernel ridge regression, we avoid the time-consuming triple-level optimization. As a result, our method is empirically 1.14 to 2.17 times faster than the previous `GCond` method.

Table A2: Comparison of running time of `GCond` and `ST-GCond`(in seconds).

| Ogbn-arxiv | r=0.05% | r=0.25% | r=0.5% |
|---|---|---|---|
| GCond (Jin et al., 2022c) | 217.18 | 386.71 | 765.12 |
| ST-GCond | 178.27 | 278.44 | 399.15 |

## E  COMPUTATION RESOURCE

We conduct all experiments with:

- Operating System: Ubuntu 20.04 LTS.
- CPU: Intel(R) Xeon(R) Platinum 8358 CPU@2.60GHz with 1TB DDR4 of Memory.
- GPU: NVIDIA Tesla A100 SMX4 with 80GB of Memory.
- Software: CUDA 10.1, Python 3.8.12, PyTorch (Paszke et al., 2019) 1.7.0.

## F  ALGORITHMS

### F.1  ALGORITHM 1

Table A3: Fixed key parameters.

| Parameters | Value |
|---|---|
| GNN backbone | GCN, GIN |
| Number of layers | 2 |
| Hidden Units | 256 |
| Activation | LeakyReLU |
| Dropout Rate | 0.5 |
| $k$ | 5 |
| Split of meta tasks | 3 |
| $\delta$ | 0.5 |

Table A4: Search space of the key parameters.

| Parameters | Search Space |
|---|---|
| **lr** | 0.1, 0.01, 0.001 |
| $\alpha$ | (0.0, 1.0) |
| $\beta$ | (0.0, 1.0) |

---

**Algorithm 1** `ST-GCond`: Self-supervised and Transferable Graph Dataset Condensation

---

1: **Input:** Graph dataset $\mathcal{G} = (\mathbf{X}, \mathbf{A}, \mathbf{Y})$, pre-trained teachers $\{f_1^{\mathcal{T}}(\cdot), \ldots, f_K^{\mathcal{T}}(\cdot)\}$, steps $T$, condensation ratio $r$, and graph sparsity parameter $\delta$.
2: **Output:** Condensed graph dataset $\mathcal{G}_s = (\mathbf{X}_s, \mathbf{A}_s, \mathbf{Y}_s^h, \mathbf{Y}_s^s)$.
3: Initialize weights $\{\lambda_i = \frac{1}{K}\}_{i=1}^K$, $\mathbf{X}_s$ by selecting $r\%$ features/class, $\mathbf{Y}_s^h$ with labels.
4: Initialize $\mathbf{A}_s = g_\phi(\mathbf{X}_s)$, $\mathbf{Y}_s^s = \frac{1}{K} \sum \lambda_i f_i^{\mathcal{T}}(\mathbf{X}_s, \mathbf{A}_s)$.
5: **for** t = $0, \cdots, T-1$ **do**
6:     Initialize $\theta \sim P_\theta$.
7:     **while** not converge **do**
8:        $D' = 0$.
9:        Sample tasks $\mathcal{T}_c \sim p(\mathcal{T}_Y)$.
10:       **for** $\mathcal{T}_i$ **do**
11:         Sample $\mathcal{G}^{\mathcal{T}_i} \sim \mathcal{G}$, $\mathcal{G}_s^{\mathcal{T}_i} \sim \mathcal{G}_s$.
12:         // *Meta-training*
13:         Adapt parameters with $\mathcal{L}_{self}$ on $\mathcal{G}_s^{\mathcal{T}_i}$: $\theta_i' \leftarrow \theta - \lambda_1 \nabla_\theta \mathcal{L}_{self}^{\mathcal{T}_i}(GNN_\theta, \mathcal{G}_s^{\mathcal{T}_i})$.
14:         // *Meta-updating*
15:         Combine representation: $\hat{\mathbf{Y}}^{\mathcal{T}_i} = \sum_{i=1}^K \lambda_i f_i^{\mathcal{T}}(\mathcal{G}^{\mathcal{T}_i})$.
16:         Compute $\mathcal{L}_{cls}, \mathcal{L}_{self}$, and $\mathcal{L}_{\text{MI}}$ on $\mathcal{G}^{\mathcal{T}_i}$:
          $D' \leftarrow D' + \left( \nabla_{\theta_i'} \mathcal{L}_{cls}^{\mathcal{T}_i}(GNN_{\theta_i'}, \mathcal{G}^{\mathcal{T}_i}) + \alpha \nabla_{\theta_i'} \mathcal{L}_{self}^{\mathcal{T}_i}(\hat{\mathbf{Y}}^{\mathcal{T}_i}, \mathcal{G}^{\mathcal{T}_i}) + \beta \nabla_{\theta_i'} \mathcal{L}_{\text{MI}}^{\mathcal{T}_i}(\mathbf{Y}_s^s; \mathbf{Y}_s^h) \right)$.
17:       **end for**
18:       Update $\{\lambda_i\}_{i=1}^K$, $\mathbf{X}_s$, $\mathbf{Y}_s^s$, $\phi$, and $\theta$.
19:     **end while**
20: **end for**
21: Generate the condensed graph: $\mathbf{A}_s = \text{ReLU}(g_\phi(\mathbf{X}_s) - \delta)$, $\mathcal{G}_s = (\mathbf{X}_s, \mathbf{A}_s, \mathbf{Y}_s^h, \mathbf{Y}_s^s)$

---

# G MORE EXPERIMENTS

## G.1 PARAMETER SENSITIVITY

We further investigate the sensitivity of the weight of $\mathcal{L}_{MI}$, which controls the influence of the mutual information loss. From Figure A1, we observe that as $\beta$ increases, the accuracy of each dataset initially rises, then fluctuates. We perform a grid search to identify the optimal setting for $\beta$. Empirically, we select $0.5, 0.6, 0.7$ based on the specific dataset.

## G.2 VISUALIZATION OF CONDENSED GRAPH

We visualize the condensed graph in the Figure A2 and report statistics in the TableA5. From the figure, we see that the learned graph is denser compared to the original, likely due to information concentration. If the condensed graph maintained the same sparsity, it might lack sufficient edges to enable effective message passing. Additionally, for the `Citeseer` and `Cora` datasets, homophily is reduced compared to the original graphs, with nodes within each class less tightly clustered. We believe it pose a more challenge to learned graph neural networks, which may have better generalization ability.

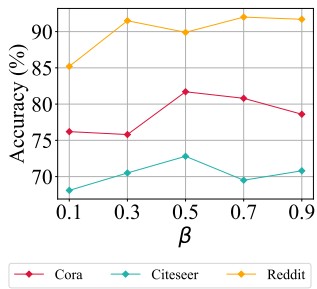

Figure A1: Sensitive of $\beta$.

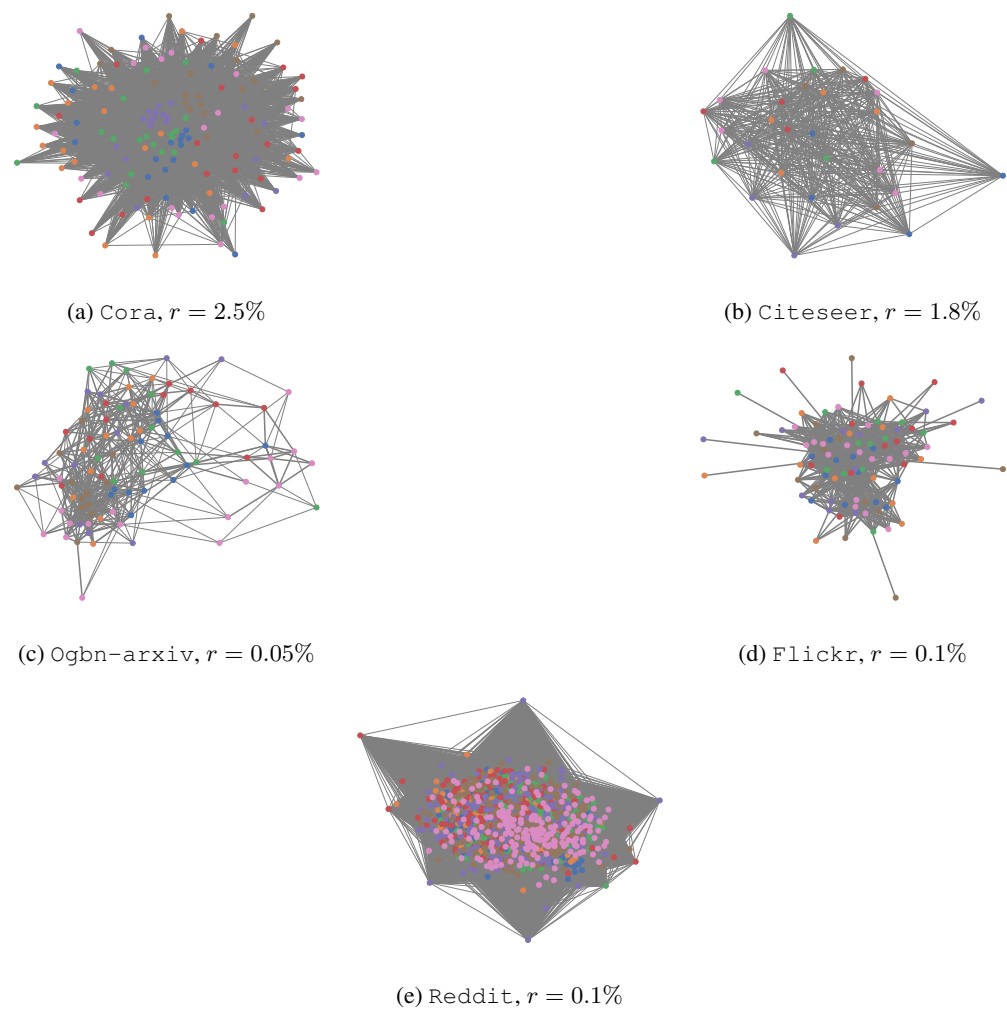

(a) `Cora`, $r = 2.5\%$

(b) `Citeseer`, $r = 1.8\%$

(c) `Ogbn-arxiv`, $r = 0.05\%$

(d) `Flickr`, $r = 0.1\%$

(e) `Reddit`, $r = 0.1\%$

Figure A2: Visualizations of the condensed graph by `ST-GCond`. The edge weights are represented by varying line thickness, and node classes are denoted by different colors.

Table A5: Statistics of the condensed graph, compared with the whole graph to highlight the differences. Note that homophily is calculated using the algorithm introduced by (Zhu et al., 2020).

| | Citeseer, r=0.9% | | Cora, r=1.3% | | Ogbn-arxiv, r=0.25% | | Flickr, r=0.1% | | Reddit, r=0.1% | |
|---|---|---|---|---|---|---|---|---|---|---|
| | Whole | ST-GCond | Whole | ST-GCond | Whole | ST-GCond | Whole | ST-GCond | Whole | ST-GCond |
| Accuracy | 70.7 | 71.5 | 81.5 | 83.4 | 71.4 | 66.8 | 47.1 | 47.5 | 93.9 | 91.7 |
| #Nodes | 3,327 | 60 | 2,708 | 70 | 169,343 | 454 | 44,625 | 44 | 153,932 | 153 |
| #Edges | 4,732 | 1,434 | 5,429 | 2,131 | 1,166,243 | 8,681 | 218,140 | 331 | 10,753,238 | 3427 |
| Sparsity | 0.09% | 77.3% | 0.15% | 84.10% | 0.01% | 8.42% | 0.02% | 34.20% | 0.09% | 29.18% |
| Homophily | 0.74 | 0.60 | 0.81 | 0.68 | 0.65 | 0.10 | 0.33 | 0.31 | 0.78 | 0.06 |
| Storage | 47.1MB | 1.1MB | 14.9MB | 0.8MB | 100.4MB | 1.3MB | 86.8MB | 0.3MB | 435.5MB | 0.7MB |

