# OpenReview forum: "ST-GCond: Self-supervised and Transferable Graph Dataset Condensation"
_ICLR.cc/2025/Conference — ICLR 2025 Poster_

### Official Review · Reviewer_z8vT · 2024-11-01

**Soundness:** 3
**Presentation:** 2
**Contribution:** 3
**Rating:** 6
**Confidence:** 4

**Summary:**

This work addresses the challenge of the transferability of condensing graph datasets in graph condensation, which is an important topic. The authors propose ST-GCond, a self-supervised and transferable graph dataset condensation method with a carefully designed loss function.

**Strengths:**

1. The transferability of condensed graphs is an important topic. The authors make the first effort to deal with it.
2. Some results are promising.
3. The motivation is clear and the writing in the introduction is good.

**Weaknesses:**

Please see the Questions below.

I'm willing to increase my rating as long as the authors can adequately address my concerns.

**Questions:**

1. I suggest the authors to provide some visualization of condensed graphs, together with some discussion on some statistical characteristics of them to make the results more credible.
2. In the Appendix D, the authors only provide and compare the running time of the proposed method together with one baseline method. Since this is an efficiency-oriented field, more comparison is needed. For example, the VRAM (GPU memory usage) during training. The proposed method introduced too many models (e.g., multiple teacher models), I'm wondering whether this will make the GPU memory usage of the proposed method even higher.
3. In cross-dataset and cross-task scenario, a finetune step on the original downstream data is needed, what if we do not undergo this step? It's weird to use the actual training data plus the condensed data to train, which leads to a downgrade of the contribution to the work .
4. What is the actual value of \alpha and \beta and other hyperparameters in the final loss function/in the main result tables? I notice the authors provide the search space of them, but did not provide the actual value / their changing trajectory during training.
5. The authors provide an ablation study on each loss terms. What's the reason for causing such results?
6. In Line 222, the author claim that "The most important step is to utilize fast adaptation in Gs." What makes it the most important step?
7. In the Appendix G.1, Line 799, the final matrix is minus by a \delta term, which comes out of no where. The authors give no explanations.

Some typos:
1. Line 396: A dot is missed at the end of the caption.
2. Line 214:  ”global minimum”  ->  "global minimum"

---

> ### Author Response · Authors · 2024-11-21
> **Overall Response to Reviewer z8vT**
>
> Dear Reviewer z8vT,
>
> We appreciate reviewer z8VT’s recognition of the importance of our research and your acknowledgment of our first pioneering efforts. We’re also glad to hear your praise for the effectiveness, motivation, and writing in the introduction. Our response to your concerns are as follow:

---

> ### Author Response · Authors · 2024-11-21
> **Response to Question #1**
>
> > **Q1:**  I suggest the authors to provide some visualization of condensed graphs, together with some discussion on some statistical characteristics of them to make the results more credible.
> >
>
> Thanks for the authors suggestion, the visualization experiments is added to the [Appendix H.2 in the revised version](https://openreview.net/pdf?id=wYWJFLQov9#page=19).

---

> ### Author Response · Authors · 2024-11-21
> **Response to Question #2**
>
> > **Q2:** I'm wondering whether this (many teacher models) will make the GPU memory usage of the proposed method even higher.
> >
>
> To address reviewers’ concern, we conduct the practical numerical comparison as follow:
>
> **Experiment setting**: we use GCond[A] and SFGC[B] as baselines to compare the preparation, condensation, and testing time, along with GPU memory usage, on the widely used Ogbn-arxiv dataset with a condensation ratio of $r=0.5\%$. All experiments are run on a single A100 GPU, and we report the overall condensation time and peak VRAM usage when the models achieve their best validation accuracy.
>
> | Method | Hyperparameters setting |
> | --- | --- |
> | GCond | inner_loop=5, outer_loop=10 |
> | SFGC | expert_epochs = 350 |
> | Ours | $k$=5, $\alpha$=0.1, $\beta$=0.6 |
> 1. Comparison on Ogbn-arxiv dataset
>
> |  |  Preparation Step (min/MB) | Condensation Step (min/MB) | Test Step (min/MB) |
> | --- | --- | --- | --- |
> | GCond | - | 251.18 / **3850.24** | 10.67 / 671.82 |
> | SFGC | 381.17 /  **1817.42** | 216.27 / 4042.18 | 10.18 /  697.33 |
> | Ours | **127.82** / 2128.21  | **207.69** / 4676.20  | 10.83 /  687.28（ $Y^h_s$ )7.82 /  759.21 ($Y^s_s$) |
>
> **Conclusion(s):**
>
> 1. **Although we introduce five additional teacher models, the increase in computational cost is modest and reletively acceptable**, with a 14.62% (310.78 MB) increase in the preparation step and a 15.6% (634.02 MB) increase in the condensation step. **We believe they are acceptable cost for the nontrivial improvements** on the cross-dataset and cross-task settings (2.5% - 18.7%).
> 2. In the condensation step, **we do not need to load whole teacher models but only the pretreained embeddings**. And we only pretrain at most 5 models in the pretrain step, which is much less than the settings of SFGC for a lot of teacher trajectories.
> 3. **For the test step, we share the similar cost on time and GPU memory with current methods, thus do not increase the burden to the downstream users.**

---

> ### Author Response · Authors · 2024-11-21
> **Response to Question #3 (Part 1/2)**
>
> > **Q3:**  In cross-dataset and cross-task scenario, a finetune step on the original downstream data is needed, what if we do not undergo this step? It’s weird to use the actual training data plus the condensed data to train, which leads to a downgrade of the contribution to the work.
> >
> 1. The finetuning step is essential for our method to mitigate the gap between the condensed dataset & task and the downstream dataset & task. **If removing this step, it equals to require the model to have the zero-shot reasoning power**, i.e., the learned model can directly inference on unseen data/task, it is highly beyond this paper’s work.
> 2. This step is also widely existed in other fields like transfer learning, meta learning, since there always exists the gap between the training dataset/task and test dataset/task. **Using the actual dataset to train the last layer or finetune the whole model is commonly used.** To further verify the practical function, **we test our method in the few-shot learning scenario**, where the actual training data only has little samples.
>
> **Baselines:**
>
> - `GPN` [C]: GPN utilize prototype networks to learn the distance from support classes to query classes.
> - `G-Meta` [D]:  G-Meta extracts the local subgraphs to mitigate the gap in generalizing to new tasks。
> - `TENT` [E]: TENT adopts complex relationship among nodes, edges, and tasks to enhance the model generalization capabilities.
>
> **Datasets and splits:**
>
> - `Children`, `Computers` : Amazon Review includes graphs composed of products, which are constructed based on co-purchased and co-viewed patterns. The Children dataset has 76,875 nodes, 1,554,578 edges, and 24 classes. The Computers dataset has 87,229 nodes, 721,081 edges, and 10 classes. We use them to test the recommender system scenario.
> - `Ogbn-arxiv` : Ogbn-arxiv is a directed citation graph between all computer science (CS) ArXiv papers indexed by Microsoft Academic Graph (MAG) . It has 169,343 nodes, 1,166,243 edges, and 40 classes. We use Ogbn-arxiv to test the citation network scenario.
>
> We begin by splitting the entire dataset into three distinct sets: the **training set, the validation set, and the test set**. Each of these sets is partitioned by class, ensuring no overlap between them. For instance, in the **ogbn-arxiv** dataset, we allocate 15 classes to the training set, 5 classes to the validation set, and the remaining 20 classes to the test set.
>
> For the test set, we apply the N-way K-shot framework typical of few-shot learning. In this setup,  we randomly select N classes from the test set. From each of these selected classes, we pick **K samples** to form the **support set**. Additionally, following TENT setting, we choose **10 samples** from the remaining unselected samples in those classes to form the **query set**, the final metric will be the average accuracy on all the query sets.
>
> We ensure consistent settings across all methods, using a train/validation/test class ratio of 3:1:4 (4:2:4 for Computers). For Ogbn-arxiv, we report results for 5-way 3-shot and 5-way 5-shot settings. For Children and Computers, we use 3-way 3-shot and 3-way 5-shot settings due to the limited number of classes.
>
> **Experiment Settings:**
>
> - For the `Linear regression` method, it only train on the **support sets** with N*K training samples, and test on the corresponding test samples on the **query sets**.
> - For the `GPN` , `G-Meta` , and `TENT` methods, they are all trained a 2-layer GCN with **training sets**, we then fixed the last layer of such GCN, and finetuned on the **support sets**. Finally, test the trained model on the **query sets**.
> - For the `ST-GCond` (Ours), we use the **training set** and their underlying graph structure to condense a small graph, then a 2-layer GCN will trained on such small condensed graph with soft and hard labels and then finetuned the last layer with **support sets.** Finally, test the trained model on the **query sets**.

---

> ### Author Response · Authors · 2024-11-21
> **Response to Question #3 (Part 2/2)**
>
> **Hyperparameters:**
>
> We utilize a 2-layer GCN as the backbone model. For the condensed graph, we apply a condensation ratio of $r = 0.5%$ for the Ogbn-arxiv dataset and $r = 1.0%$ for the Children and Computers datasets. Regarding other hyperparameters for the baseline models, we adopt the optimal settings as reported in their respective papers.
>
> |  | Children |  | Computers |  | Ogbn-arxiv |  |
> | --- | --- | --- | --- | --- | --- | --- |
> |  | 3-way 3-shot | 3-way 5-shot | 3-way 3-shot | 3-way 5-shot | 5-way 3-shot | 5-way 5-shot |
> | Linear regression | 39.68±0.61 | 43.33±1.78 | 37.19±1.36 | 40.11±1.72 | 53.24±0.77 | 59.31±0.71 |
> | GPN | 54.77±0.42 | 60.51±1.16 | 71.73±1.67 | 70.47±1.72 | 61.65±0.67 | 66.34±0.78 |
> | G-Meta | 54.00±2.38 | 57.76±1.43 | 71.58±2.52 | 71.14±2.52 | 60.06±1.56 | 63.77±2.53 |
> | TENT | 53.23±0.76 | 60.52±1.73 | 62.13±1.17 | 62.13±1.17 | 62.24±0.35 | 65.80±0.37 |
> | GCond | 51.23±1.17 | 52.82±0.78 | 61.28±1.29 | 61.23±2.01 | 58.32±1.27 | 60.28±1.02 |
> | Ours | **61.27**±1.27 | **63.18**±0.89 | **74.56**±0.99 | **73.28**±1.23 | **65.72**±1.23 | **70.11**±1.65 |
>
> **Conclusion(s):**
>
> 1. The traditional graph condensation method `GCond` **performs poorly in this setting**, outperforming only simple `Linear regression` while falling behind meta-learning baselines. In contrast, our approach surpasses all runner-up methods by a margin of 2.9% to 10.6%, demonstrating **its practical utility in few-shot scenarios** and its effectiveness in enhancing the **cross-task generalization capabilities** of condensed graphs.
> 2. Compared to our manuscript, we use the entire target task/dataset as the “support set” and ”query set” to ensure a fair comparison with the **whole dataset accuracy** reported by traditional graph condensation methods. **We appreciate the reviewer’s suggestion and will include detailed few-shot scenario experiments in a future revision to highlight practical value.** Thank you for pointing this out.

---

> ### Author Response · Authors · 2024-11-21
> **Response to Question #4**
>
> > **Q4:**  What is the actual value of \alpha and \beta and other hyperparameters in the final loss function/in the main result tables?
> >
>
> Thank you for the suggestion. We provide the detailed configuration here and include it in Appendix F of the revised version.
>
> 1. Key parameters of ST-GCond:
>
>
>     | Parameters | Value |
>     | --- | --- |
>     | GNN backbone | GCN (Node level) GIN (Graph level) |
>     | Number of layers | 2 |
>     | Hidden Units | 256 |
>     | Activation | LeakyReLU |
>     | Dropout Rate | 0.5 |
>     | Number of teachers ($k$) | 5 |
> 2. Detailed configuration for $lr$, $\alpha$, $\beta$, $k$
>
> For the Table 1. Node classification performance comparison under the single task and dataset scenario
>
> |  | $r$ | $lr$ | $\alpha$ | $\beta$ |
> | --- | --- | --- | --- | --- |
> | Citeseer | 0.90% | 0.001 | 0.7 | 0.5 |
> |  | 1.80% | 0.001 | 0.7 | 0.5 |
> |  | 3.60% | 0.001 | 0.7 | 0.7 |
> | Cora | 1.30% | 0.001 | 0.7 | 0.5 |
> |  | 2.60% | 0.001 | 0.7 | 0.5 |
> |  | 5.20% | 0.001 | 0.7 | 0.5 |
> | Ogbn-arxiv | 0.05% | 0.01 | 0.5 | 0.6 |
> |  | 0.25% | 0.01 | 0.7 | 0.6 |
> |  | 0.50% | 0.01 | 0.5 | 0.6 |
> | Flickr | 0.10% | 0.001 | 0.5 | 0.7 |
> |  | 0.50% | 0.001 | 0.5 | 0.6 |
> |  | 1.00% | 0.001 | 0.7 | 0.8 |
> | Reddit | 0.05% | 0.01 | 0.5 | 0.8 |
> |  | 0.10% | 0.01 | 0.4 | 0.8 |
> |  | 0.20% | 0.01 | 0.6 | 0.8 |
>
> For the Table 2. Node classification performance (Accuracy% ± std) comparison under the cross-dataset scenario
>
> |  | $lr$ | $\alpha$ | $\beta$ |
> | --- | --- | --- | --- |
> | Ogbn-arxiv → target datasets | 0.01 | 0.8 | 0.6 |
>
> For the Table 3. Graph classification performance (ROC-AUC(%) ± std) comparison under the cross-dataset scenario
>
> |  | $lr$ | $\alpha$ | $\beta$ |
> | --- | --- | --- | --- |
> | GEOM → target datasets | 0.01 | 0.5 | 0.7 |
>
> For the Table 4. Link prediction results under the cross-task scenario
>
> |  | $lr$ | $\alpha$ | $\beta$ |
> | --- | --- | --- | --- |
> | Cora | 0.001 | 0.6 | 0.7 |
> | Citeseer | 0.001 | 0.7 | 0.5 |
>
> For the Table 5. Node clustering results under the cross-task scenario
>
> |  | $lr$ | $\alpha$ | $\beta$ |
> | --- | --- | --- | --- |
> | Cora | 0.001 | 0.4 | 0.8 |
> | Citeseer | 0.001 | 0.6 | 0.7 |

---

> ### Author Response · Authors · 2024-11-21
> **Response to Question #5**
>
> > **Q5:**  The authors provide an ablation study on each loss terms. What's the reason for causing such results?
> >
> 1. **Result 1:** All variants perform significantly better than the baselines. **Reason:** This improvement may be attributed to the naive methods are narrowly focused on task-specific information, which can hinder downstream learning. In contrast, our two component is designed to simulate cross-task and cross-dataset scenarios, helping to bridge the gap between the condensed dataset and the downstream test dataset.
> 2. **Result 2:** The naive joint learning approach using both hard and soft labels (ST-GCond w/o MI) performs worse than the other two variants, ST-GCond w/o self and ST-GCond w/o meta. **Reason:** This may be due to conflicts arising from the jointly use of supervised and self-supervised tasks. Similar discussions can be found in pretraining studies, which also motivate our design of a mutual information component to mitigate these conflicts.

---

> ### Author Response · Authors · 2024-11-21
> **Response to Question #6**
>
> > **Q6:**  In Line 222, the author claim that "The most important step is to utilize fast adaptation in Gs." What makes it the most important step?
> >
>
> In response of “Efficient and Fast Cross-task Adaptation”, we propose the “Task-distangled Meta Optimization” strategy. Briefly, inspired by MAML[E], we want to simulating the scenario where users update their models on new data and test on the updated version. **The** **most important step** denotes how to simulate and improving $G_s$’s performance in such scenario.
>
> By revisiting the manuscript, we find such short statement may causing misunderstanding for the reader, **we revised the “the most important step” to “the key step” and rewrite the expression as follow, we use the blue to highlight them in the revised paper.**
>
> ```latex
> The key step is simulating the cross-task scenario in the condensing stage with $\mathcal{G}_s$.
> ```

---

> ### Author Response · Authors · 2024-11-21
> **Response to Question #7**
>
> > **Q7:**  In the Appendix G.1, Line 799, the final matrix is minus by a \delta term, which comes out of no where. The authors give no explanations.
> >
>
> The $\delta$ here denotes the **graph sparsity parameter** following the algorithm 1 of GCond[A], we briefly introduce it in the line 219-221. Thanks for pointing out, we revise our expression as follows. We also revise the Appendix G.1 in the revision, and add the $\delta = 0.5$ to the Appendix F parameter setting table.
>
> ```latex
> \textbf{Initialization of the Condensed Graph $\mathcal{G}_s$.} We initialize $\{\mathbf{X}_s, \mathbf{Y}_s^h\}$ by randomly selecting a subset of the original data. Following \texttt{GCond}, we use an MLP $g_\phi$ as the structure generator, where $\mathbf{A}_s = (g_\phi(\mathbf{X}_s) - \delta)$, **Here, $\delta$ serves as the sparsity parameter to filter out edges with lower weights**. Note that $\mathbf{Y}_s^h$ is analogous to the labels $\mathbf{Y} \in \mathbb{R}^{N \times 1}$ of $\mathcal{G}$, encompassing $C$ classes.
> ```

---

> ### Author Response · Authors · 2024-11-21
> **Conclusion**
>
> **We also correct the typos in our revised paper** and sincerely thank the reviewers for your time and valuable feedback. We hope our clarifications and additional few-shot learning results address your concerns. **If the revisions meet your expectations, we respectfully request raising of the scores, it can greatly help us.**
>
> Best,
>
> Authors from Submission 2264.
>
> ---
> **Reference**
>
> [A] Wei Jin et.al. Graph condensation for graph neural networks. ICLR 2022
>
> [B] Xin Zheng et.al. Structure-Free Graph Condensation: From Large-Scale Graphs to Condensed Graph-Free Data. NeurIPS 2024
>
> [C] Kaize Ding et.al. Graph prototypical networks for few-shot learning on attributed networks.
> CIKM 2020.
>
> [D] Kexin Huang et.al. Graph meta learning via local subgraphs. NeurIPS 2020.
>
> [E] Song Wang et.al. Task-adaptive few-shot node classification. KDD 2022.
>
> [F]  Chelsea Finn et.al. Model-agnostic meta-learning for fast adaptation of
> deep networks. ICML 2017.

---

> > ### Comment · Reviewer_z8vT · 2024-11-21
> >
> > I appreciate the authors' efforts in addressing my comments. Most of my concerns have been adequately resolved, and I have raised my score from 5 to 6. However, I remain concerned about the trade-off between efficiency and utility in this method. This is super important in such efficiency-driven field. I recommend that the AC and PC consider this aspect when making their decisions.

---

> > > ### Author Response · Authors · 2024-11-25
> > >
> > > Dear reviewer z8vT,
> > >
> > > We hope this message finds you well.
> > >
> > > We sincerely thank you once again for your insightful comments and quick responses. We are pleased to see that our efforts, including additional visualizations and few-shot experiments, have addressed most of your concerns!
> > >
> > > As the author-reviewer discussion period is nearing its conclusion, we kindly ask if our further clarifications on time complexity and practical numerical comparisons **have helped resolve your remaining concerns** regarding the efficiency and utility trade-off of our method? If not, we are more than happy to provide further explanations or experiments.
> > >
> > > We truly value your feedback and remain committed to addressing any remaining concerns!
> > >
> > > Best regards,
> > >
> > > Authors from Submission 2264.

---

> ### Author Response · Authors · 2024-11-22
>
> Dear reviewer z8vT,
>
> We are pleased that the reviewer acknowledges our efforts in addressing most of your concerns and has raised the score accordingly! We still want to address all of your concern during the public period, we provide detailed responses as follows:
>
> **1. Our method shares the similar time complexity with current methods, and only introduce ~15% more GPU usage to get up to 18.7% improvements, which we think is acceptable.**
>
> Regarding the efficiency-utility trade-off, we want to clarify that our method retains a comparable time complexity to existing approaches (proof provided in Appendix TODO and detailed comparisons in **response to Q1 by reviewer 8R18**). Additionally, based on empirical results, our peak GPU usage is in line with baseline methods, **as shown in our response to Q2**. For convenience, we summarize the combined comparison here. Note we omit the SGDD due to its high computational cost of optimal transport ($O(N’^{2.373})$), and GEOM that introduce addtional burden of curriculum Learning comparing to SFGC.
>
>
> | Methods | Time complexity |  Preparation Step (min/MB) | Condensation Step (min/MB) | Test Step (min/MB) |
> | --- | --- | --- | --- | --- |
> | GCond | $TKO(Lmd^2 + Lmd) + TKO(m^2d^2)$ | - | 251.18 / **3850.24** | 10.67 / 671.82 |
> | SFGC | $MTO(Lmd^2 + Lmd) + TO(r^LNd^2)$ | 381.17 /  **1817.42** | 216.27 / 4042.18 | 10.18 /  697.33 |
> | Ours | $(t+1)TO(r^LNd^2 + m^2d^2 + Nmr^2) + TkO(LNd^2)$ | **127.82** / 2128.21  | **207.69** / 4676.20  | 10.83 /  687.28（ $Y^h_s$ )7.82 /  759.21 ($Y^s_s$) |
>
> **Conclusion(s):**
>
> - **We share the same order of time complexity with existing methods**: Our approach maintains a similar order of complexity, $O(Nd^2)$ and $O(md^2)$, with potentially smaller constant factors.
> - **We achieve acceptable trade-off on GPU usage and performance**: While introducing five additional teacher models, the computational cost increase is modest—14.62% (310.78 MB) for preparation and 15.6% (634.02 MB) for condensation. This trade-off yields notable performance improvements of 2.5%–18.7% in cross-dataset and cross-task scenarios, making it a worthwhile balance.
>
> **2. We believe that efficiency should prioritize the testing phase over the condensation step**, **as the ultimate goal is to benefit downstream usage effectively.**
> - *Condensation step*: **current approaches are inherently inefficient and computationally expensive.** For instance, gradient descent-based methods often require **multiple model reinitializations**, while trajectory-matching methods involve **tens or even hundreds of single training iterations** to generate teacher trajectories, making the condensation step resource-intensive and time-consuming.
> - *Test step***:** In contrast, all condensed graph is highly efficient for downstream applications, requiring less than 10 minutes and under 1000 MB of GPU memory. This aligns perfectly with the core purpose of graph dataset condensation: **to provide a compact, plug-and-play graph for a wide range of downstream tasks.** This process can be compared to training large language models (LLMs), where enormous computational resources and money are required to train the model, but the resulting weights can later be efficiently utilized even on a modest 16 GB CPU-only laptop for various usage. **Though graph condensation methods cannot achieve LLM’s impact, we share the same underlying motivation, and kindly request the tolerance from reviewers.**
>
> We sincerely thank the reviewer for their insightful comments and **we really hope to address all the reviewer’s concerns during the public discussion period.** Additionally, we view this feedback as a valuable opportunity to guide our future work, thereby achieving our goal of enhancing the overall impact of research within this topic.
>
> Best,
>
> Authors from Submission 2264.

---

### Official Review · Reviewer_8R18 · 2024-11-02

**Soundness:** 3
**Presentation:** 3
**Contribution:** 2
**Rating:** 8
**Confidence:** 3

**Summary:**

This paper presents a novel graph dataset condensation method termed ST-GCond that enhances transferability across tasks and datasets using a multi-teacher self-supervised optimization strategy. ST-GCond effectively condenses large graph datasets, which can maintain high performance in varied applications by leveraging multiple pre-trained models and self-supervised learning. Experiments demonstrate ST-GCond's effectiveness in both single-task/single-dataset and cross-task/cross-dataset scenarios.

**Strengths:**

ST-GCond performs well across different tasks and datasets and overcomes the limitations in traditional graph condensation methods.

By using multiple pre-trained models as teachers, the proposed method captures a wide range of features and knowledge, and thereby improving its ability to generalize.

**Weaknesses:**

The proposed ST-GCond method introduces substantial computational overhead due to task-disentangled meta optimization and multi-teacher self-supervised optimization. Given the iterative updates for each sub-task and multiple self-supervised models, the overall training time could be significantly increased. It would be beneficial if the authors could provide detailed analysis of computational cost .

The proposed method would benefit from a thorough exploration of hyperparameter tuning, such as the number of sub-tasks and learning rates.  Including a sensitivity analysis or providing guidelines on hyperparameter selection based on different types of graph datasets could be beneficial.

Significant disparities between tasks may limit the condensed graph's ability to capture all relevant information, and thereby reducing the effectiveness.

The power of the multi-teacher strategy relies on the quality of pre-trained models. Will low-quality or irrelevant models impair the performance?

The effectiveness of the multi-teacher self-supervised optimization strongly hinges on the optimal configuration of the weights  assigned to each teacher model's output. However, this paper does not provide a clear illustration for how these weights are optimized during the training process.

**Questions:**

Can you provide a detailed analysis of the computational costs and compare them with conventional methods?

What strategies are used for tuning hyperparameters like the number of sub-tasks and learning rates? Could you include guidelines for hyperparameter selection?

How does the quality of pre-trained models affect the performance of ST-GCond?

Can you explain how the weights assigned to each teacher model's output are optimized during training?

---

> ### Author Response · Authors · 2024-11-21
> **Overall Response to Reviewer 8R18**
>
> Dear Reviewer 8R18,
>
> We are glad to hear that you recognize the novelty of our method in overcoming the limitations of traditional graph condensation approaches, and  we appreciate your praise for the carefully designed framework we’ve developed to capture a broad range of knowledge. Our response to your questions are as follow:

---

> ### Author Response · Authors · 2024-11-21
> **Response to Question #1**
>
> > **W1 & Q1:**  Can you provide a detailed analysis of the computational costs and compare them with conventional methods?
> >
>
> **We provide the time complexity of our method in Appendix D of the manuscript.** To facilitate clearer discussions with the reviewers, we slightly revise the notation for better alignment and comparison with the baselines.
>
> ST-GCond primarily consists of two parts: 1. Sampling sub-tasks for meta-updating. 2. Involving self-supervised tasks to guide condensing.
>
> - **Sampling Sub-Tasks**
>
> For the first part, we treat the sampling as a composition of bi-level optimization. Following GCond, we use an $L$-layer GCN where The large-scale graph has $N$ nodes. The condensed graph has $m$ nodes. The hidden dimension is $d$. The edge of the original graph is $E$, the number of the sampled neighbors per node be $r$.
>
> The computation cost per task includes: Forward pass through the original GNN: $O(r^LNd^2)$. Forward pass through $g_\phi$: $O(m^2d^2)$. Inner optimization of kernel ridge regression: $O(Nmr^2 + Nm)$.
>
> Thus, the single-task complexity is: $O(r^LNd^2 + m^2d^2 + Nmr^2 + Nm).$ For $t$ split tasks, the total complexity for this part is: $tO(r^LNd^2 + m^2d^2 + Nmr^2 + Nm).$
>
> - **Invloving self-supervised tasks**
>
> In the condensing stage, multiple self-supervised models are introduced. Using an **offline strategy**, the additional computation complexity is: $kO(LEd + LNd^2),$ where $k$ denotes the number of self-supervised tasks.
>
> - **Overall Complexity**
>
> The overall complexity of ST-GCond is expressed as: $(t+1)O(r^LNd^2 + m^2d^2 + Nmr^2 + Nm) + kO(LEd + LNd^2).$  With ignore the lower-order terms in the expression, we get:
>
> $(t+1)O(r^LNd^2 + m^2d^2 + Nmr^2) + kO(LNd^2).$
>
> Compare with other methods: we further define the $T$ as the iteration, $K$ as the times of initialization, $N’$ as the sample of the original graph, $M$ as the number of teachers. The comparison is shown as follow:
>
> | GCond | $TKO(Lmd^2 + Lmd) + TKO(m^2d^2)$ |
> | --- | --- |
> | SGDD | $TKO(Lmd^2 + Lmd) + TKO(m^2d^2) + TKO(N'^{2.373})$  |
> | SFGC | $MTO(Lmd^2 + Lmd) + TO(r^LNd^2)$  |
> | GEOM | $MTO(Lmd^2 + Lmd) + TO(r^LNd^2)$  |
> | Ours | $(t+1)TO(r^LNd^2 + m^2d^2 + Nmr^2) + TkO(LNd^2).$ |
>
> **Conclusion(s):**
>
> 1. The order of complexity is similar to other methods with $md^2$ (since $m^2$ can be considered of the same order as $m$ because $m \ll N$) and $Nd^2$. For the factor $t$ and $k$, we set it to 3 and 5 in most of our experiments, which is comparable to or lower than $K$ and $M$. **Therefore, our time complexity is comparable to other methods.**
> 2. The second term, $TkO(LNd^2)$, corresponds to the embedding of the original graphs and can be precomputed to improve efficiency. To support this, we provide time and GPU memory comparisons in response to Q2 from reviewer z8vT. **The results demonstrate that our method is more efficient than SFGC in the preparation step, as it avoids the need to precompute numerous teacher models. During the condensation stage, our approach performs comparably to other methods.**

---

> ### Author Response · Authors · 2024-11-21
> **Response to Question #2**
>
> > **W2 & Q2:** What strategies are used for tuning hyperparameters like the number of sub-tasks and learning rates? Could you include guidelines for hyperparameter selection?
> >
> - **number of sub-tasks:** it doesn’t affect the performance a lot according to our empirical study, so we set to **3** in our experiments.
> - **learning rates**, we use the fixed search space on $\{0.1, 0.01, 0.001\}$, we recommend to try the $0.01,0.001$ first, since they are show good performance on our empirical studies (see our response to the Q4 to the reviewer z8vT or our revised Appendix F.)
> - **self-supervised term’s weight $\alpha$ and MI term’s weight** $\beta$: our observation is that the performance varies with different values. Based on our empirical setting, we suggest to use the $\{0.5, 0.6, 0.7, 0.8\}$ searching space for such two values.
> - **other key parameters like layers, hidden units, number of teachers:** please see our response to the Q4 to the reviewer z8vT and our revised appendix F.

---

> ### Author Response · Authors · 2024-11-21
> **Response to Question #3**
>
> > **W3, W4 & Q3:** How does the quality of pre-trained models affect the performance of ST-GCond?
> >
>
> To fairly compare the quality of pre-trained models, we use the experiments to show the quality impact.
>
> **Experiment Setup:** We use the Cora dataset in a single-task and single-dataset scenario with condensation ratio $r=5.2\%$. In this setup, we condense the Cora graph and employ the condensed graph to train a downstream GCN) for evaluating its performance on the Cora dataset, we also conduct 5 test to align with the current results on our manuscript.
>
> To evaluate the effectiveness of the teacher model, we modify the pretraining process by varying the number of epochs according to specific ratios of the original pretraining durations. For instance, in our manuscript, we use 200 epochs for pretraining the CLU method. By applying a 10% ratio, we reduce the pretraining time to just 20 epochs.
>
> 1. **Only use 1 model to serve as the teacher models:**
>
> | pretraining epoch ratio: | 10% | 30% | 50% | 70% | 90% | 100% |
> | --- | --- | --- | --- | --- | --- | --- |
> | DGI | 80.8±0.2 | 81.2±0.2 | 81.7±0.4 | 82.3±0.1 | 82.3±0.1 | 82.1±0.1 |
> | CLU | 80.3±0.1 | 80.8±0.5 | 81.6±1.2 | 81.4±0.3 | 82.1±0.4 | 82.6±0.3 |
> | PAR | 73.3±1.2 | 72.3±2.1 | 73.3±1.6 | 75.8±0.9 | 77.3±0.6 | 77.8±0.8 |
> | PAIRSIM | 76.5±1.8 | 72.4±1.2 | 75.9±0.4 | 76.2±0.8 | 78.6±0.4 | 79.0±0.3 |
> | PAIRDIS | 69.5±2.5 | 72.1±2.1 | 73.2±1.8 | 75.8±1.2 | 76.4±1.3 | 78.3±0.6 |
> 1. **Fix 4 models, change the 1 model’s quality:**
>
> | pretraining epoch ratio: | 10% | 30% | 50% | 70% | 90% | 100% |
> | --- | --- | --- | --- | --- | --- | --- |
> | DGI | 82.0±0.6 | 82.3±0.8 | 82.6±0.3 | 83.0±0.7 | 82.9±0.2 | 83.6±0.9 |
> | CLU | 82.3±1.0 | 82.5±0.3 | 82.5±0.3 | 82.8±0.8 | 83.3±0.6 | 83.6±0.9 |
> | PAR | 83.1±0.8 | 83.3±0.4 | 83.3±0.3 | 83.6±0.3 | 83.6±0.3 | 83.6±0.9 |
> | PAIRSIM | 82.7±0.4 | 82.8±0.3 | 83.0±0.4 | 83.3±0.4 | 83.3±0.4 | 83.6±0.9 |
> | PAIRDIS | 82.8±0.5 | 83.1±0.4 | 83.5±0.6 | 83.4±0.3 | 83.5±0.6 | 83.6±0.9 |
>
> **Conclusion(s):**
>
> 1. **Improved pretraining quality leads to better performance, but the effect varies by method.** As the pretrained models improve (more pretraining epochs), both settings show increased performance. However, we find that poor-quality targets can hinder the condensation process, as seen with the 10% pretraining case, where results are worse than using hard labels alone (81.8%).
> 2. **Our adaptive multi-teacher framework remains effective even with one model quality fluctuations.** From the second setting, we observe that changing one model out of five still produces relatively strong results compared to using a single model, demonstrating the robustness of our method.

---

> ### Author Response · Authors · 2024-11-21
> **Response to Question #4**
>
> > **W5 & Q4:** Can you explain how the weights assigned to each teacher model's output are optimized during training?
> >
>
> We define the variables $\{\lambda_i\}_{i=1}^k$ to control the weight of each teacher model. **During end-to-end training, these weights are optimized via backpropagation**. Initially, the weights undergo significant changes and gradually converge over time. The changes in the weights are illustrated for the Cora and Citeseer datasets in Fig. 4(a) and Fig. 4(b) of our manuscript.

---

> ### Author Response · Authors · 2024-11-21
> **Conclusion**
>
> We thank for the reviewer to point out the efficiency and pretrain model quality problems, we hope our responses and additional experiments address your concern. **We kindly request you to raising the score accordingly if you satisfied with our response, it means a lot to us!**
>
> Best,
>
> Authors from Submission 2264.

---

> ### Author Response · Authors · 2024-11-22
>
> Dear Reviewer 8R18,
>
> With the public discussion phase coming to a close soon, we want to check if our responses have sufficiently addressed your comments and concerns. Your insights are invaluable to us, and we’re eager to use this opportunity to enhance our work.
>
> Thank you once again for your thoughtful feedback and contributions.
>
> Best regards,
>
> Authors of Submission 2264.

---

> ### Author Response · Authors · 2024-11-25
> **Summary of Rebuttal**
>
> Dear Reviewer 8R18,
>
> We hope this message finds you well.
>
> Apologies for emailing you again. We kindly remind you that the author-reviewer discussion period will close in less than two days. We sincerely want to know if we **have address all your concerns** and would greatly appreciate any opportunity to **further clarify or respond**. Below is a summary of our responses to your questions for your convenience:
>
> - **Q1:** *Can you provide a detailed analysis of the computational costs and compare them with conventional methods?* **A1:** We compare the **time complexity of our method with existing approaches**. The results show that our time complexity is **comparable** to other methods and is even **more efficient** than SFGC during the **preparation step (**train and save teacher models’ trajectory).
> - **Q2:** *What strategies are used for tuning hyperparameters like the number of sub-tasks and learning rates? Could you include guidelines for hyperparameter selection?* **A2:** We provide **guidelines for each hyperparameter** and include detailed configurations in the revised **Appendix F**.
> - **Q3:** *How does the quality of pre-trained models affect the performance of ST-GCond?* **A3:** We conducted **additional experiments**, varying the training epochs of pre-trained models and testing settings with 1 model and 5 models. Our findings show that **improved pretraining quality leads to better performance, but the effect varies by method.** Nonetheless, our **adaptive multi-teacher framework remains robust even with fluctuations in the quality of a single model.**
> - **Q4:** *Can you explain how the weights assigned to each teacher model’s output are optimized during training?* **A4:** The weights assigned to each teacher model’s output are optimized through **end-to-end training using backpropagation.**
>
> Once again, we sincerely thank the reviewer for their time and effort in reviewing our manuscript. We firmly believe that this discussion will greatly help strengthen our paper!
>
> Best regards,
>
> Authors from Submission 2264.

---

> ### Author Response · Authors · 2024-12-01
>
> Dear reviewer 8R18,
>
> We sincerely thank reviewer 8R18 for raising the score to 8 in support of our work! Your feedback is greatly appreciated, and we are committed to addressing any outstanding questions.
>
> Best regards,
>
> Authors from Submission 2264.

---

### Official Review · Reviewer_AjtW · 2024-11-04

**Soundness:** 3
**Presentation:** 2
**Contribution:** 3
**Rating:** 6
**Confidence:** 3

**Summary:**

This manuscript studies graph dataset condensation from a different perspective. It proposes a method for condensing the graph dataset in a cross-dataset and cross-task manner. The proposed ST-GCond condenses the dataset while preserving the most universal/general information, which is task-agnostic.  Specifically, there are two components of the method. The first is task-disentangled meta optimization which makes the condensed dataset aware of the task difference. The second is multi-teacher self-supervised optimization which makes the dataset hold some uniserval information. Experiments and ablation studies are well-done with nontrivial performance gain.

**Strengths:**

- It is great to see that the proposed method improves under the setup even in the single dataset and task. The performance gain under the setup of cross-dataset and cross-task is indeed nontrival.

- The proposed method is applicable for both node-level and graph-level tasks, making it more general.

- The proposed method makes it work with the combination of multi-task learning and self-supervised learning on the dataset condensation, which might inspire more researchers on this research topic.

**Weaknesses:**

- For the "Mutual Information Guided Joint Condensation", it is unclear why the performance is dropped when we use both the hard label from the supervised condensation and the soft label from the self-supervised condensation. The author argues that this is due to the conflict

- The cross-task and cross-dataset dataset condensation settings are indeed interesting and are more applicable in real scenarios. However, such settings are a little bit overlap with that of the graph model pertaining. The pertaining of the model can also achieve faster adaption to the new task or dataset. Can the authors provide a more detailed discussion of the differences?

- From the ablation studies, it is shown that the proposed method can even achieve the best performance with either only the "self" part or the "meta" part. Why do the authors say that "ST-GCond w/o self and ST-GCond w/o meta perform poorly on both datasets"?

- It is a little bit unfair for the comparison as the proposed method utilizes much more information during the dataset condensation. For example the multi-label information (for the meta-training part) and self-supervised models (for the self-training part).

- What if we do not have the multi-task information for the dataset we would like to condense?

**Questions:**

Please refer to the weaknesses part.

---

> ### Author Response · Authors · 2024-11-21
> **Overall Response to Reviewer AjtW**
>
> Dear Reviewer AjtW,
>
> We sincerely appreciate your recognition of our nontrivial performance across both cross-dataset and cross-task scenarios, as well as our framework’s applicability to both node-level and graph-level tasks. We also hope our work serves as a source of inspiration for future research! Our responses to your questions are outlined below.

---

> ### Author Response · Authors · 2024-11-21
> **Response to Weakness #1**
>
> > **W1:** Why does performance drop when using both hard and soft labels?
> >
>
> This observation is from our empirical study of the ST-GCond variant without $\mathcal{L}_{MI}$ in the ablation analysis. In this case, we use the mean of all teachers’ output embeddings as the sole target when jointly training with soft and hard labels. Unexpectedly, its performance is even worse than directly using the averaged teachers’ embeddings for testing. A specific example is shown below.
>
> **Experiment setting:** We conducted an ablation study on our framework under the cross-dataset setting. Specifically, we condensed a dataset from Ogbn-arxiv and generalized it to the Cora dataset (as outlined in Table 1 of our paper). The ablated components include:
>
> - Soft labels can be produced using a mutual-information-guided approach or without guidance, relying solely on the mean average of all teacher models‘ output.
> - While hard labels are initialized at the start, their corresponding optimization of condensed graph can be performed either with or without incorporating the multi-task meta-learning mechanism.
>
> | Settings | (1) | (2) | (3) | (4) | (5) |
> | --- | --- | --- | --- | --- | --- |
> | $\mathbf{Y}^s_s$  | w $\mathcal{L}_{MI}$ | w $\mathcal{L}_{MI}$ | w/o $\mathcal{L}_{MI}$ (mean average) | w/o $\mathcal{L}_{MI}$ (mean average) | w/o $\mathcal{L}_{MI}$ (mean average) |
> | $\mathbf{Y}^h_s$ | w $\mathcal{L}_{cls}^{meta}$ | w/o $\mathcal{L}_{cls}^{meta}$ | w $\mathcal{L}_{cls}^{meta}$ | w/o $\mathcal{L}_{cls}^{meta}$ | None (only use the initialized condensed graph) |
> |  | 81.1 | 76.4 | **64.7** | **73.9** | **71.1** |
>
> The lowest performance is observed when combining the mean-averaged outputs of teacher models with the meta-learning paradigm **(Setting 3)**. This setup underperforms compared to scenarios without meta-learning **(Setting 4)** or even without optimizing the graph structures **(Setting 5)**, **highlighting challenges in joint optimization**. As discussed in prior knowledge distillation studies [A, B], this performance drop likely stems from conflicts in optimization directions. **Specifically, the self-supervised optimization objectives may misalign with those of supervised optimization during joint training, leading to suboptimal outcomes.**

---

> ### Author Response · Authors · 2024-11-21
> **Response to Weakness #2**
>
> > **W2:** Can the authors provide a more detailed discussion on the differences between the setting in this paper and that of a pretrained model?
> >
>
> This is an important point in distinguishing our work, and we sincerely thank the reviewers for highlighting it. In the paper, we briefly address the difference in the introduction (lines 51-53), and here we offer a more comprehensive explanation of this distinction:
>
> - **Motivation disparity:** Our primary goal is to extend existing graph condensation methods to further benefit the **low computational resource situation**, as it is highly impractical for users in such scenario to rely exclusively on a condensed graph tailored to a single, limited dataset and task. In contrast, graph pretraining methods do not emphasis computational constraints; they typically provide **source code and the original dataset, expecting users to either reimplement the method or directly incorporate it into downstream end-to-end workflows,** which is impossible for our target low computational resource situation.
> - **More flexible solution:** For those graph pretraining methods who offer out-of-the-box model weights with predefined architectures to enhance downstream applications, **they still face the limitation in meeting the diverse requirements of downstream users.** For instance, in low computational resource environments, users may need fewer hidden units and more efficient convolutional backbones. Below, we summarize the out-of-the-box weights from current graph pretrained methods which provide weights:
>
> | Methods | GPT-GNN [C] (#option num) | pretrain-gnns [D] (#option num) | GraphMVP [E](#option num) | Ours (#option num) |
> | --- | --- | --- | --- | --- |
> | hidden-layers | 1 (3-layer) | 1 (5-layer) | 1 (5-layer) | No limitation |
> | hidden-dimension | 1 (400) | 1 (300) | 1 (300) | No limitation |
> | activation | 1  | 1 | 1 | No limitation |
> | backbones | 5 (HGT, GCN, GAT, RGCN, and HAN) | 4 (GIN, GCN, GAT, and GraphSAGE) | 1 (GIN) | No limitation |
> | Summary | 5 | 5 | 1 | No limitation |
>
> Here are a few examples; however, in practice, most well-known methods still require users to start from scratch. **In contrast, our approach provides flexibility by offering only the dataset, eliminating the need for fixed downstream architecture specifications.**
>
> - **Compatibility with existing models:** Our approach is also plug-and-play, seamlessly integrating with pre-existing models. **For instance, in continual learning scenario, our method can support model updates with the condensed graph.** In contrast, pretrained weights are typically applicable only during the initial model-building phase.
> - **Open to customized needs:** In previous pretraining studies addressing domain-specific challenges (e.g., social networks, citation networks, biological graphs), researchers typically choose a pretrained model from the most relevant domain or rely on large, redundant models to handle multiple domains. **In contrast, our approach provides greater flexibility by allowing users to select datasets and easily combine them, enabling customized training tailored to specific downstream tasks.**
>
> To the best of our knowledge, this work is the first to introduce graph condensation methods into cross-task and cross-dataset scenarios, aiming to expand the range of options available for downstream users. **We will incorporate  a systematic comparison and discussion in the revised version as suggested.** Furthermore, we are delighted for further discussion with the reviewers on this point.

---

> ### Author Response · Authors · 2024-11-21
> **Response to Weakness #3**
>
> > **W3:**  the ablation study, why the authors says that two variants perform poorly on both datasets?
> >
>
> **It is because it still worse than the naive GCN results on whole dataset.** Our intention was to convey that these two variants are not as effective as the combined approach. While both variants outperform the baselines in cross-task and cross-dataset scenarios, **they are still too imprecise compared to the full dataset in real-world applications.** For example, the whole Cora dataset should have 81.2% accuracy, but the variant ST-GCond w/o meta only have 76.4%, while the ST-GCond has the 81.5%.
>
> **Apologies for the confusion in the manuscript, we refine the wording as follows:**
>
> ```latex
> As observed, ST-GCond w/o self and ST-GCond w/o meta demonstrate strong performance relative to the baselines, likely due to the shortcomings of naive methods, which tend to overemphasize task-specific information. However, both variants still fall short when compared to the results obtained using the full dataset (e.g., accuracies of 81.2\% on Cora and 93.9\% on Reddit), emphasizing the necessity of combining these components for optimal performance.
> Nevertheless, the naive combination of such two parts ST-GCond w/o MI underperforms both individual variants, . Similar observations have been discussed in prior works [D, E], further highlighting the critical role of $\mathcal{L}_MI$ in resolving such conflicts and improving overall functionality.
> ```

---

> ### Author Response · Authors · 2024-11-21
> **Response to Weakness #4**
>
> > **W4:** Unfair comparison for utilizing much more information.
> >
>
> Thanks for the comment. **We do not think that our method use much more information than previous methods**. While we introduced several techniques to empower our condensed graph to generalize across datasets and tasks, we still maintain the same information and settings during the condensation phase as earlier methods. Our detailed response is provided below:
>
> - **We share the same amount of information with reduction/condensation baselines under a fair setting.** As summarized in the following table, for task-specific information, our multi-task meta learning framwork just better simulates cross-task scenarios while utilizing the same amount of information as existing methods. Regarding universal information, often tied to graph structure, prior methods like herding and SGDD also leverage it in reduction. **Our approach just offers a more effective mechanism for extracting such knowledge, but share the same amount of information.**
>
> |  | Random | Herding | K-Center | GCond | SFGC | GEOM | SGDD | Ours |
> | --- | --- | --- | --- | --- | --- | --- | --- | --- |
> | Task information | X | X | X | √ | √ | √ | √ | √ |
> | Universal information | X | √ | √ | X | X | X | √ | √ |
> - **Our method does not use the downstream dataset and task information during training, which has much less information comparing to naive GCN baseline.** To the best of our knowledge, this is the first work to extend graph condensation to cross-dataset and cross-task scenarios, with a primary focus on surpassing the naive GCN baseline (see Fig. 1(c) and Fig. 1(d)) across entire target datasets and tasks. Without such capability, the method would be impractical for real-world applications. **However, since downstream dataset and task information is unavailable during the condensation process, we believe our comparison represents a fair and realistic setting for practical use cases.**

---

> ### Author Response · Authors · 2024-11-21
> **Response to Weakness #5**
>
> > **W5:** What if we do not have the multi-task information for the dataset we would like to condense.
> >
>
> **We can disentangle the current task into a multi-task scenario.** In the Task-Disentangled Meta Optimization step, the given task (e.g., classes) can typically be broken down into multiple subsets, each containing more than one class. It is also a widely-used method in the meta-learning methods [F].
>
> **We can still get considerable performance under task information unavailable scenario.** In extreme case where task label information is unavailable (no any task labels), actually, **all of current condensed methods generally fail to perform  condensation. However, we can still rely solely on the Multi-Teacher Self-Supervised Optimization module to generate target node embeddings for initializing the condensed graph and soft labels.**
>
> **Experiment setting:** In extreme cases where task label information is unavailable, we can still initialize the condensed graph using the outputs of the self-supervised teacher models as soft labels. The other components of the condensed graph are summarized in the following table:
>
> |  | $\mathbf{X}_s$ | $\mathbf{A}_s$ | $\mathbf{Y}^h_s$ | $\mathbf{Y}^s_s$ |
> | --- | --- | --- | --- | --- |
> | Multi-task information | Optimized | Optimized | Assigned | Optimized |
> | Task information unavailable | Select from original graph | Constructed by Cosine pairwise similarity | - | Select from multi-teacher generated embeddings |
>
> In the cross-dataset and cross-task scenarios, we test the condensed graph from task information unavailable setting, where the `Baselines` denotes the MLP (w/o pre) and VGAE methods.
>
> |  | Ogbn-arxiv → Cora ($r$ = 0.50% , Accuracy%) | Node Clustering on Cora ($r$ = 0.5%, F1%) |
> | --- | --- | --- |
> | Baselines (MLP w/o pre and VGAE) | **54.8** | **57.5** |
> | Ours (Multi-task information) | 81.1 | 71.4 |
> | Ours (Task information unavailable) | **71.1** | **64.8** |
>
> We observe that the condensed graph, even without task-specific information, still outperforms the baselines. This advantage may attribute to **its equivalence to distilling the downstream model using the output embeddings generated by the multi-teacher framework.**

---

> ### Author Response · Authors · 2024-11-21
> **Conclusion**
>
> Once again, we thank the reviewer for your insightful comment and warmly invite them for  further discussion. **If we addressed all your concerns, we kindly hope that you will consider raising score accordingly.**
>
>
> Best,
>
> Authors from Submission 2264.
>
> ---
>
> **Reference**
>
> [A] Wei Jin et.al. Automated self-supervised learning for graphs. ICLR 2022.
>
> [B] Tianyu Fan et.al. Decoupling weighing and selecting for integrating multiple graph pre-training tasks. ICLR 2024.
>
> [C] Ziniu Hu et.al. GPT-GNN: Generative Pre-Training of Graph Neural Networks. KDD 2020.
>
> [D]  Weihua Hu et.al. Strategies for Pre-training Graph Neural Networks. ICLR 2020.
>
> [E] Shengchao liu et.al.  Pre-training Molecular Graph Representation with 3D Geometry. ICLR 2022.
>
> [F] Kexin Huang et.al. Graph meta learning via local subgraphs. NeurIPS 2020.

---

> ### Author Response · Authors · 2024-11-22
>
> Dear reviewer AjtW,
>
>
> As the public discussion phase is nearing its conclusion, we want to kindly follow up to confirm if our responses have addressed all your concerns. Please don’t hesitate to share any additional questions or feedback. We believe this would be an excellent opportunity to further refine our manuscript!
>
>
> Thank you once again for your valuable input.
>
>
> Best regards,
>
> Authors of Submission 2264.

---

> ### Author Response · Authors · 2024-11-25
> **Summary of Rebuttal**
>
> Dear Reviewer AjtW,
>
> We hope this message finds you well.
>
> Apologies for bothering you again. As the author-reviewer discussion period will conclude in two days, we provide a brief summary of our responses to your questions for your convenience:
>
> - **W1:** Why does performance drop when using both hard and soft labels?
> **A1:** We addressed this through an **additional ablation study**, which revealed that the self-supervised optimization objectives may **misalign** with those of supervised optimization during joint training, leading to suboptimal outcomes.
> - **W2:** Can the authors provide a more detailed discussion on the differences between the setting in this paper and that of a pretrained model?
> **A2:** We elaborated on this from four aspects: **Motivational disparity**, **More flexible solutions**, **Compatibility with existing models**, and **Openness to customized needs**. We appreciate the reviewer for highlighting this point.
> - **W3:** the ablation study, why the authors says that two variants perform poorly on both datasets?
> **A3:**  Our comparison here is against the original **whole dataset accuracy**. To prevent further confusion, we revised the manuscript for clarity.
> - **W4:** Unfair comparison for utilizing much more information.
> **A4:** We compare the amount of information we used with baseline methods, we share the **same amount** with existing methods, and use much **less information** comparing to the naive GCN method.
> - **W5:** What if we do not have the multi-task information for the dataset we would like to condense.
> **A5:** Using our **task-disentangled technique**, we can simulate a “multi-task” scenario. Furthermore, we tested an **extreme case with** **no task information** and demonstrated that our method **remains effective** using only the self-supervised optimization module.
>
> We would greatly appreciate it if the reviewer could let us know **whether we have adequately addressed the reviewers’ concerns**. Additionally, please let us know if there are any **further clarifications or additional questions** that we can address.
>
> Best regards,
>
> Authors from Submission 2264.

---

> ### Author Response · Authors · 2024-12-01
>
> Dear Reviewer AjtW,
>
> Thank you once again for your insightful review. We believe that answering your questions will significantly improve the quality of our manuscript.
>
> As the extended discussion period approaches its conclusion (ending in 48 hours), we are eager to know if our responses adequately resolve your concerns. If not, we would be happy to provide further feedback or clarifications during the remaining time.
>
> Best regards,
>
> Authors from Submission 2264.

---

### Author Response · Authors · 2024-11-21
**Global Response**

#

We sincerely thank all reviewers for their detailed and constructive feedback on our manuscript. We are delighted that they  acknowledge our pioneering contributions and recognize the effectiveness of our method. Specifically, reviewer acknowledged that:

1. **Pioneering contributions:** Reviewer `AjtW` believes the combination of multi-task learning and self-supervised learning **could inspire further research**. Reviewer `8R18` commends our work for **overcoming the limitations of traditional methods**, while reviewer `z8vT` acknowledges the **importance of our research topic** and recognizes our **first effort** in dealing it.
2. **Effective method:** Reviewer `AjtW` acknowledges the **nontrivial** improvement in our method’s generalization to **both graph-level and node-level tasks**. Reviewer `8R18` commends our approach for **capturing a wide range of knowledge**, which improving generalization. Reviewer `z8vT` finds our results **promising**.

During the rebuttal period, we provide **visualizations of our condensed graph**, **detailed hyper-parameter configurations**, **detailed comparison on time complexity and GPU usage**, **additional few-shot learning results**, and further clarifications aimed at addressing the reviewers’ concerns. We also summarize some of these updates and uploaded the [revised manuscript](https://openreview.net/pdf?id=wYWJFLQov9) for further consideration.

Once again, we would like to express our gratitude to all the reviewers for their time and effort in evaluating our paper. Their thoughtful comments have significantly contributed to improving the quality and potential impact of our work. After carefully revising the paper based on their valuable feedback, we hope we address all concerns and kindly request the reviewers to consider raising their scores accordingly.

Best,

Authors from Submission 2264.

---

### Meta-Review · Area_Chair_QpYs · 2024-12-11

**Metareview:**

This paper studies graph dataset condensation from a new perspective, which is meaningful and interesting.  To be specific, existing methods require downstream usage to match the original dataset and task, which is impractical in many real-world scenarios. In contrast, this paper shows that existing methods fail in "cross-task" and "cross-dataset" scenarios, often performing worse than training from scratch. To address these challenges, the authors propose a novel method termed Self-supervised and Transferable Graph dataset Condensation (ST-GCond). For cross-task transferability, the authors propose a task-disentangled meta optimization strategy to adaptively update the condensed graph according to the task relevance, encouraging information preservation for various tasks. For cross-dataset transferability, they propose a multi-teacher self-supervised optimization strategy to incorporate auxiliary self-supervised tasks to inject universal knowledge into the condensed graph.

Generally speaking, the contributions of this paper have been recognized by the reviewers. The experimental results also verify the effectiveness of the proposed method when compared with existing methods.

**Additional Comments On Reviewer Discussion:**

The reviewers generally show positive scores to this paper. The authors are also actively involved in the rebuttal process, which is helpful in addressing reviewers' concerns. In the discussion phase, the reviewers do not have further concerns. Therefore, I recommend an acceptance to this paper.

---

### Decision · Program_Chairs · 2025-01-22

Accept (Poster)